# Quantitative optical nanoscopy of mitochondrial-derived vesicles in neurons classifies pre-peroxisomal and clearing organelles

Giovanna Coceano [1], Jonatan Alvelid [1,2], Martina Damenti [1], Gabriella Ferretti [1,3], Johannes Mueller[1], Joanna Rorbach [4] & Ilaria Testa [1]

Healthy mitochondria are crucial for maintaining neuronal homeostasis. Their activity depends on a dynamic lipid and protein exchange through fusion, fission, and vesicular trafficking. Studying vesicles in neurons is challenging with conventional microscopy due to their small size, heterogeneity, and dynamics. We use multicolour stimulated emission depletion nanoscopy to uncover the ultrastructure of mitochondrial-derived vesicles (MDVs) in live neurons, biosensors to define their functional state, and a pulse-chase strategy to identify their turnover in situ. We identified three populations of vesicular structures: one transporting degradation products originating from oxidative stress, one shuttling cargo and newly translated proteins for local organelle biogenesis and one consisting of small, functional mitochondria. Furthermore, we provide evidence supporting that de novo peroxisomes biogenesis occurs via the fusion of endoplasmic reticulum and MDVs at mitochondrial sites. Our data provide mechanistic insight into organelle biogenesis driven by significant diversity in MDV morphology, functional state, and molecular composition.

Mitochondria are highly dynamic structures that continually alter their morphology through fusion and division to ensure mitochondrial homeostasis and support cellular functions[1-3]. In addition to fusion and fission, recent research has revealed key mechanisms governing mitochondrial membrane plasticity. These mechanisms involve either direct membrane contact or vesicular transport to facilitate the exchange of lipids and proteins between mitochondria and the surrounding environment[4-6]. These processes are particularly significant in neurons, where the extended and polarized cellular shape poses additional challenges in coordinating the distribution of organelles and nutrients across various compartments[7]. Among the different pathways, mitochondrial-derived vesicles (MDVs) have emerged as key players in mitochondrial signaling within the cell, participating in a diverse array of functions[8]. MDVs play a pivotal role in mitochondrial quality control (MQC) by transporting oxidized cargo that is targeted for degradation within the endolysosomal system[9-12]. This process operates under basal conditions and increases during oxidative stress when damage is localized and does not extend to the entire organelle[13-15]. Ultrastructural analysis via electron microscopy has revealed that MDVs are characterized by either single or double membranes[16]. Fluorescence microscopy studies have elucidated their diverse protein cargo, suggesting different functions including selective transport of mitochondrial cargo to peroxisomes[17,18], antigen presentation on the plasma membrane[19,20], and packaging of

[1]Department of Applied Physics and SciLifeLab, KTH Royal Institute of Technology, Stockholm, Sweden. [2]Department of Biophysical Imaging, Leibniz Institute of Photonic Technology, Jena, Germany. [3]Unit of Pharmacology, Department of Neuroscience, School of Medicine, University of Naples Federico II, Naples, Italy. [4]Department of Medical Biochemistry and Biophysics, Karolinska Institutet, Stockholm, Sweden. ✉e-mail: ilaria.testa@scilifelab.se

mitochondrial proteins in extracellular vesicles[21]. Despite extensive studies of MDVs in immortalized cell lines and other cell types, their formation, dynamics, and functional roles in neurons remain poorly understood. Neurons are highly polarized and compartmentalized, with axons and dendrites exhibiting distinct morphology, protein composition, and functional requirements. Notably, in axons, a specialized class of vesicles removes the mitochondrion-anchoring protein Syntaphilin, facilitating mitochondrial motility[22], highlighting the potential for compartment-specific regulation of mitochondrial dynamics. These observations suggest the possibility that MDVs in neurons are heterogeneous, differing in biogenesis, cargo composition, and function depending on their subcellular location. Investigating MDVs in primary neurons is therefore critical to understanding how these vesicles contribute to compartment-specific MQC, organelle turnover, and overall neuronal homeostasis—mechanistic insights that cannot be fully captured in standard cell line models. Moreover, while trafficking of mitochondrial proteins and lipids is essential for cellular homeostasis, the precise function, composition, and subcellular localization of MDVs remain largely undefined, particularly in neurons. This ambiguity may stem from the dense clustering of mitochondria and MDVs within confined spaces like axons and synapses, which poses a challenge for accurate visualization using conventional fluorescence microscopy techniques. To directly observe specific proteins associated with MDVs in live neuronal cells, we employed stimulated emission depletion (STED) nanoscopy in conjunction with bright cell-permeable dyes[23] and genetically encoded self-labeling enzymes (SNAP/Halo)[24,25]. This approach offers several advantages: firstly, multicolor STED provides the spatial resolution necessary to observe and quantify the fine morphology of single MDVs within the intricate neuronal architecture, enabling us to resolve the budding-out process from mitochondria; and secondly, by employing a diverse set of sensors, we can assess the functional state of MDVs in situ. As the imaging is performed in living neurons, we additionally implemented a sequential labeling strategy to track the lifetime and trafficking of the proteins in the MDVs. Our findings reveal MDVs exhibiting spherical and tubular shapes of different sizes, distributed across various neuronal compartments. These MDVs contain either degradation products resulting from oxidative stress or newly synthesized proteins enriched in peroxisomal biogenesis markers. Our imaging data show the presence of endoplasmic reticulum (ER) at MDV formation sites, also for peroxisomal-positive MDVs, as well as the presence of Dynamin-related protein 1 (DRP1), supporting the hypothesis that fission machinery is required for the biogenesis of pre-peroxisomal vesicles. Altogether, we shed light on ultrastructural features of MDVs and elucidate their diverse role in the clearance of mitochondrial debris and peroxisome biogenesis within primary neurons.

## Results

### MDS heterogeneity and distinct turnover dynamics in neurons

We applied STED nanoscopy to fixed and live primary rat hippocampal neurons to visualize the morphology of mitochondria in different cellular compartments during neuronal development (DIV 5–10) (Fig. 1 and Supplementary Fig. 1). The C-terminal localization peptide of the mitochondrion´s outer membrane protein 25 (OMM) was fused to either the self-labeling Halo/SNAP-tags (Halo/SNAP-OMM) or the rsEGFP2 protein[26,27](rsEGFP2-OMM), for live- or fixed- mitochondria labeling, respectively. With this strategy, all outer mitochondrial membranes were labeled and compatible with STED imaging[23,28], preserving mitochondrial morphology[29], and reducing off-target labeling effects[30] (Fig. 1a, b and Supplementary Fig. 1a–d). The mitochondrial morphology was quantified by three main parameters: area ($A$), width ($W$), and length ($L$) (Supplementary Fig. 1a), which were extracted by our custom analysis pipeline "Mitography" (Supplementary Fig. 1i). Analyzing nearly 1400 mitochondria in STED images, mitochondrial

area, width, and length values were significantly smaller than those obtained by confocal imaging (Fig. 1a inset and Supplementary Fig. 1e, f). Thus, STED but not confocal microscopy could resolve individual mitochondria (Supplementary Fig. 1c, d) and define their vesicular and tubular fine structures. Mitochondrial width values were distributed narrowly around a mode of 139 nm, with 75% of mitochondria having a width of 100–200 nm. However, mitochondria with widths below 100 nm (3.2%) and above 300 nm (5.1%) were also present. Instead, the width distribution obtained by confocal microscopy displayed a mode of 357 nm, with 80% of values above 300 nm and an overall wider distribution. Similarly, mitochondrial length values obtained with STED were shorter compared to confocal analysis, even for length values well above the diffraction limit. In fact, while confocal imaging showed elongated streaky shapes that could be misinterpreted as a single long mitochondrion, STED imaging would resolve individual mitochondria aligned in series along axons and dendrites (Supplementary Fig. 1d). Similar trends were also true for the measured area, where STED, but not confocal, analysis revealed a significant population smaller than 0.086 μm$^2$ ($A_{th}$) (Fig. 1a inset). We therefore established an area threshold at this value to focus on these fine structures, defined here as mitochondrial-derived structures (MDSs). At this point, this may include MDVs, fragments, and small functional mitochondria. In the subsequent experiments, we combined well-established MDV markers with precise measurements of size and morphology to characterize the different subsets of MDSs present in neurons, with specific attention to MDVs. MDSs were as small as 80 nm across, and they were often located near larger mitochondria, making it difficult to identify them in confocal imaging (Fig. 1b). The distribution of the area was broad, with a mode of 0.035 μm$^2$, corresponding, for example, to a length and width of 233 nm and 150 nm, respectively (Fig. 1c and Supplementary Fig. 1g, h). By calculating the aspect ratio (AR) of MDSs as the ratio between length and width, AR = $W/L$, we could investigate their ultrastructure heterogeneity further (Fig. 1d). Circular MDSs were defined as having an AR of 0.5–1, while AR < 0.5 represents tubular MDSs, denoted as sticks (Fig. 1e). We further investigated the spatial distribution of vesicles and sticks across the neuronal cell, and we found that both MDS types were equally distributed in axons and dendrites (Fig. 1f).

Profiting from the spatio-temporal resolution of STED nanoscopy, we followed the formation of MDSs in living neurons over time, confirming that these structures indeed originated from mitochondria (Supplementary Movies 1–3). We tracked events such as fission followed by the additional formation of a vesicle with a diameter of approximately 90 nm at the site of fission (Supplementary Fig. 1j and Supplementary Movie 1). Furthermore, we observed MDSs budding out from lateral protrusions of the mitochondrial membrane through the formation of a thin constriction (vesicle neck, 72 nm), followed by the excision of a newly formed vesicle (Supplementary Fig. 1k, l and Supplementary Movie 2). Interestingly, at sites of vesicle formation, we observed the ER, suggesting possible similarities with the mechanism of mitochondrial fission (Supplementary Fig. 1m, n and Supplementary Movie 3)[31].

Considering the observed variety in sizes and shapes and literature data about the involvement of MDVs in mitochondrial protein trafficking and recycling, we checked whether mitochondria and MDSs were associated with different turnover dynamics. We applied a sequential labeling method based on the self-labeling SNAP-tag. In this experiment, mitochondrial membrane replacement over time and space can be detected by imaging the color variation in the images (Fig. 2). Neurons were transfected with a SNAP-tag plasmid fused to the C-terminal localization peptide of OMP25 (SNAP-OMM). After 24 h ($t = 0$), we pulsed the cells for 1 h with the red fluorescent ligand tetramethylrhodamine (TMR-Star-BG) to label all mitochondrial membranes tagged with SNAP-OMM at time zero ($t = 0$). After multiple washing steps, the culture was sent back to the incubator. Twenty-four hours later ($t = 24$ h), we pulsed the cells once more with the ligand

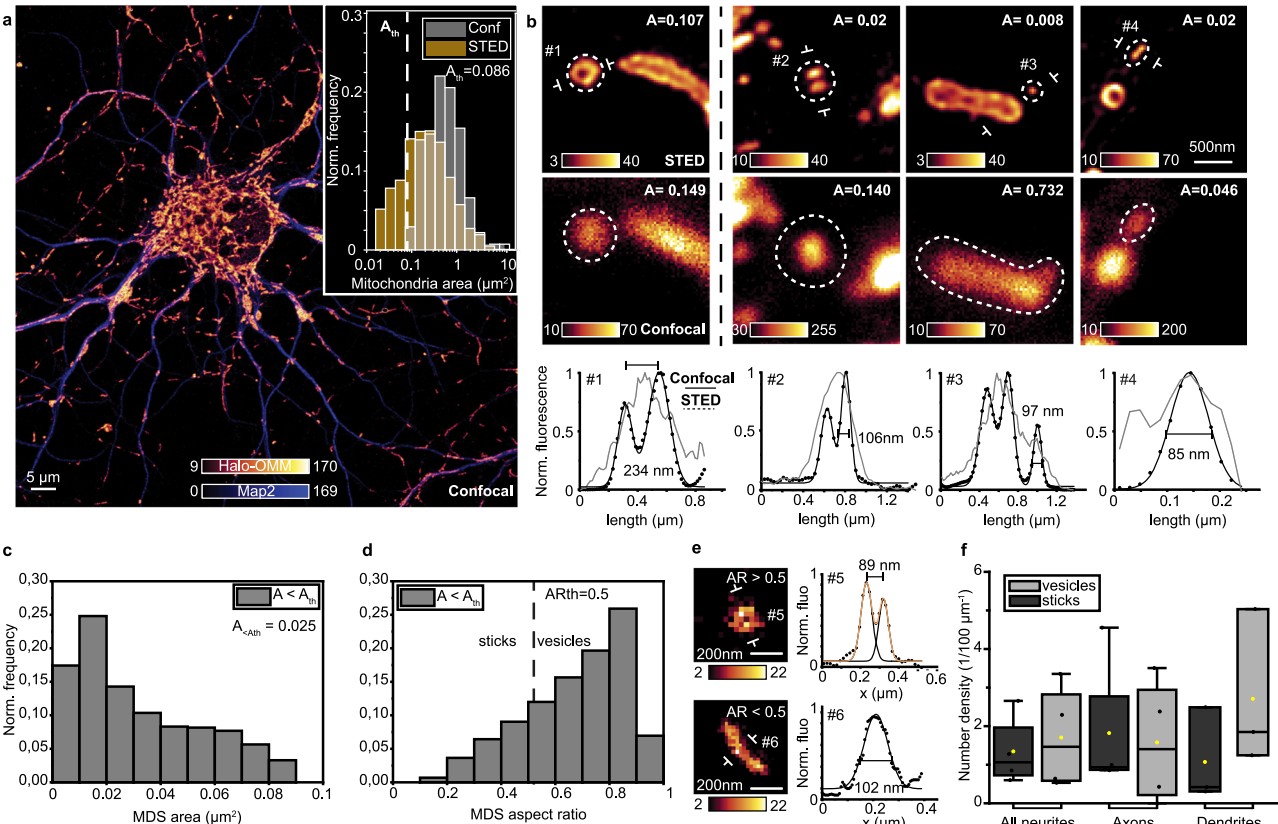

**Fig. 1 | MDS heterogeneity in neurons. a** Confocal image of the neuronal mitochondrial network, stained through the C-terminal localization peptide of the mitochondrial outer membrane protein OMP25 (OMM) fused to the Halo-tag (Halo-OMM, red hot) and the dendritic filaments, immunostained for the microtubule-associated protein 2 (MAP2, blue). The inset shows mitochondrial area distribution as measured in STED (yellow) or confocal (gray); the dashed line shows the threshold value ($A_{th}$ = 0.086 μm²), chosen to discriminate between mitochondria and mitochondrial-derived structures (MDSs). Data were collected from 31 DIV7–9 hippocampal neurons from more than three independent experiments. $N_{STED}$ = 1391 organelles, $N_{Confocal}$ = 136 organelles. **b** STED (top) and confocal (middle) images of mitochondria ($A > A_{th}$) and MDSs ($A < A_{th}$), as representative examples of the data plotted in (**c**, **d**). Normalized intensity line profiles from indicated lines (bottom), as measured in confocal (gray line) and STED (black dots), and Gaussian fits of STED data (black line). **c** MDS area distribution as a histogram

with bin width 0.01 μm². Data were collected from 21 DIV7–9 hippocampal neurons from more than three independent experiments. $N_{MDVs}$ = 637, median = 0.025 μm². **d** MDS aspect ratio (AR) distribution as a histogram with bin width 0.1 μm². Data were collected from 21 DIV7–9 hippocampal neurons from four independent cultures. The dotted line shows the threshold value $A_{th}$ discriminating between sticks (AR < $A_{th}$) and vesicles (AR > $A_{th}$). $N_{MDVs}$ = 637. **e** STED images (left) and normalized line profiles (right) of representative vesicular and tubular structures, with FWHM of 89 nm and 102 nm, respectively (dots are data points; lines are Gaussian fits). The images are representative examples of the data plotted in (**f**). **f** Number density distribution of MDSs across different neuronal compartments. Data are expressed as mean ± standard deviation. No statistically significant differences have been observed. Data were collected from 25 FOVs taken from four cells from two independent experiments. Student's t-tests: $P_{all\ neurites}$ = 0.68; $P_{axons}$ = 0.85; $P_{dendrites}$ = 0.30.

coupled with the far-red silicon rhodamine dye (SiR-647-BG) (Fig. 2a). By imaging the cells after the second pulse, we could differentiate between old and newly produced outer mitochondrial membranes, labeled with two spectrally distinct dyes. We found that the population was highly heterogeneous in content with newly formed and old vesicles containing either one label or the other, but the majority showed colocalization of the two dyes, highlighting a continuous membrane turnover occurring within 24 h (Fig. 2b–e). To quantify the turnover rate, we measured the fluorescence intensity ratio between the newly labeled and the total amount of proteins, i.e., the sum of old and new. Taking into account that exogenous SNAP-OMM was integrated both in the soma and in the distal filaments already within the first 4 h of protein expression and that its distribution remained constant within 24 h (Supplementary Fig. 2a–d), we found that the turnover did not happen at the same rate in each compartment; in fact, both mitochondria and MDSs refreshed their membrane content faster near the cell soma than in distal filaments, as measured by the progressive decrease in the intensity ratio between the SiR-647-BG labeling and the total labeling intensity at increasing distances from the cell soma (Fig. 2f).

In conclusion, STED nanoscopy revealed significant diversity in the size, shape, and turnover of MDSs, with faster protein turnover for MDSs located near the cell soma than for those in distal filaments.

## MDS are characterized by different functional states

Considering the variety of shapes and sizes observed with STED and the numerous functions to which MDVs have been associated with, we wanted to investigate whether sizes might be related to specific functions. MDVs have been described as a possible mechanism for MQC[12], working together with mitophagy and other pathways[32] to clear mitochondria from potentially damaging products in different cell types. The formation of such vesicles is triggered by mitochondrial stress, and they are enriched in oxidized proteins[16]. We used STED imaging in combination with functional readouts to elucidate the physiological state of super-resolved MDSs in neuronal cells to identify MDVs involved in MQC. Mitochondria were labeled, as previously described, to visualize the overall population, with the addition of a spectrally separated signal generated from two markers of mitochondrial state: (i) tetramethylrhodamine ethyl ester (TMRE), to stain the fully functional and metabolically active mitochondria, or (ii)

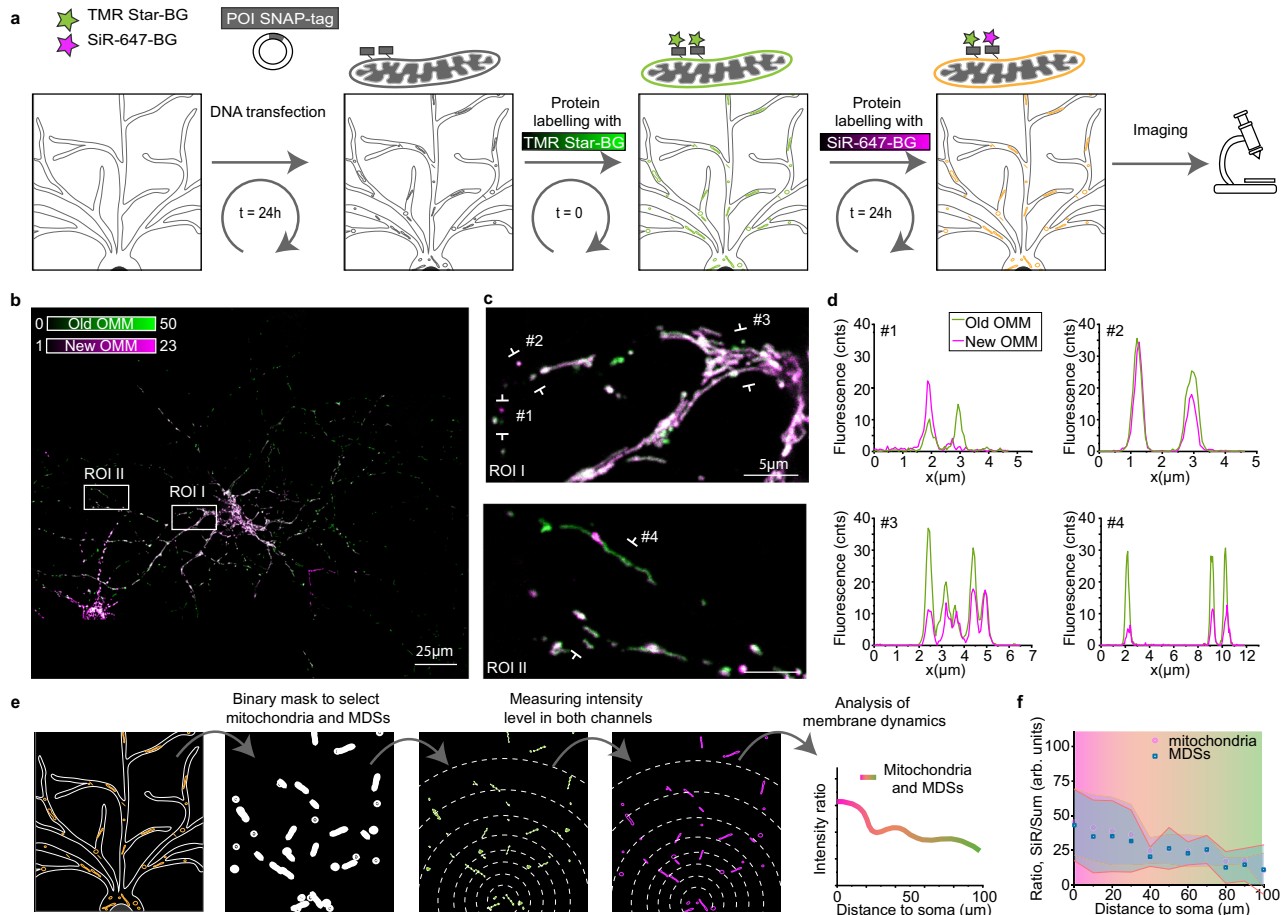

**Fig. 2 | MDSs are associated with different turnover dynamics. a** Workflow scheme of the SNAP-tag protein expression system to follow membrane turnover at different time points with fluorescence microscopy. SNAP-tag allows the labeling of the protein of interest (POI) with SNAP-BG ligands functionalized with two different dyes, TMR-Star-BG (green) and SiR-647-BG (magenta). **b** Tile-scanning confocal image (5 individual square tiles, each one 100×100μm2) of a neuronal cell where membrane refreshment, labeled with SNAP-OMM, can be followed at different time points thanks to the sequential labeling of old ($t = 0$ h, old OMM, green) and new ($t = 24$ h, new OMM, magenta) mitochondrial membranes. The image is a representative example of the data plotted in (**f**). **c** Two representative regions of interest (ROIs) from (**b**) showing mitochondria in proximity to the cell soma (ROI I) and in distal filaments (ROI II). **d** Intensity line profiles from the indicated signs in (**c**), highlighting the heterogeneity in lifetime of MDSs at different distances from the cell soma. **e** Workflow scheme of the image analysis for the quantification of membrane refreshment across the neuronal cell. **f** Quantification of the ratio between SiR-647-BG (new OMM) and the sum of SiR-647-BG and TMR-Star-BG intensities in mitochondria (pink dots) and MDSs (blue squares), measured at different distances from the cell soma. Data are expressed as mean (dots) ± standard deviation (shaded area). Data were collected from five cells from two different experiments.

MitoSOX, to highlight mitochondria undergoing oxidative stress (Fig. 3). The majority (73%) of mitochondria were labeled by TMRE and these were larger as compared to those with very weak signal (mean area: $A_{TMRE+} = 0.423\,\mu m^2$, $A_{TMRE-} = 0.069\,\mu m^2$) (Fig. 3a, d). Similar trends were observed for MDSs ($A < A_{th}$, $N = 105$): TMRE⁺ were larger (mean area: $A_{TMRE+} = 0.062\,\mu m^2$, $A_{TMRE-} = 0.040\,\mu m^2$) and longer (mean length: $L_{TMRE+} = 0.36\,\mu m$, $L_{TMRE-} = 0.28\,\mu m$) than TMRE⁻ (Fig. 3b, e). Most MDSs were round-shaped vesicles, of which 68% lacked membrane potential. Similar results were obtained for the sticks (71% with no membrane potential) (Fig. 3f, g), suggesting the possible role of these membranous vesicles without membrane potential in MQC. STED illumination had minimal impact on mitochondrial membrane potential and morphology, as neurons showed similar loss of TMRE signal and mitochondrial stability under both STED and confocal imaging conditions (Supplementary Fig. 3a–d and Supplementary Note 1).

Next, we investigated the presence of reactive oxygen species (ROS) inside MDSs using MitoSOX (Fig. 3c). Most mitochondria were labeled by MitoSOX (82%), indicating, as expected, the presence of ROS. Mitochondria without MitoSOX were on average smaller than those with (mean area: $A_{MitoSOX-} = 0.10\,\mu m^2$, $A_{MitoSOX+} = 0.32\,\mu m^2$,

Fig. 3h). Focusing on MDSs, the majority were MitoSOX⁺ (69%) and were similar in size (mean area: $A_{mitoSOX+} = 0.045\,\mu m^2$, $A_{mitoSOX-} = 0.039\,\mu m^2$) (Fig. 3i) but had different shapes compared to the MitoSOX⁻ MDVs. Vesicles were enriched in ROS, while sticks were not (mean aspect ratio: $AR_{mitoSOX+} = 0.70$, $AR_{mitoSOX-} = 0.63$) (Fig. 3j, k). MitoSOX alone did not affect mitochondrial ROS production, as its fluorescence remained stable unless combined with phototoxic MitotrackerGreen, which increased ROS through photo-bleaching (Supplementary Fig. 3h, i and Supplementary Note 1).

Together, these experiments showed that neuronal MDSs are highly heterogeneous in size and function, suggesting a possible relation between MDSs and MDVs. We observed small MDSs ($A = 0.04\,\mu m^2$; median length and width: $L = 250$ nm, $W = 160$ nm) lacking membrane potential but showing ROS accumulation. These vesicles have characteristics similar to previously reported small mitochondria and MDVs that accumulate oxidized proteins and are subject to autophagosome engulfment[3,10,16], pointing to a potential role in MQC. Instead, larger MDSs ($A = 0.06\,\mu m^2$) have the same features as mitochondria, having both membrane potential and ROS.

To better understand the nature of these vesicles and to differentiate them from small functional mitochondria, we labeled the

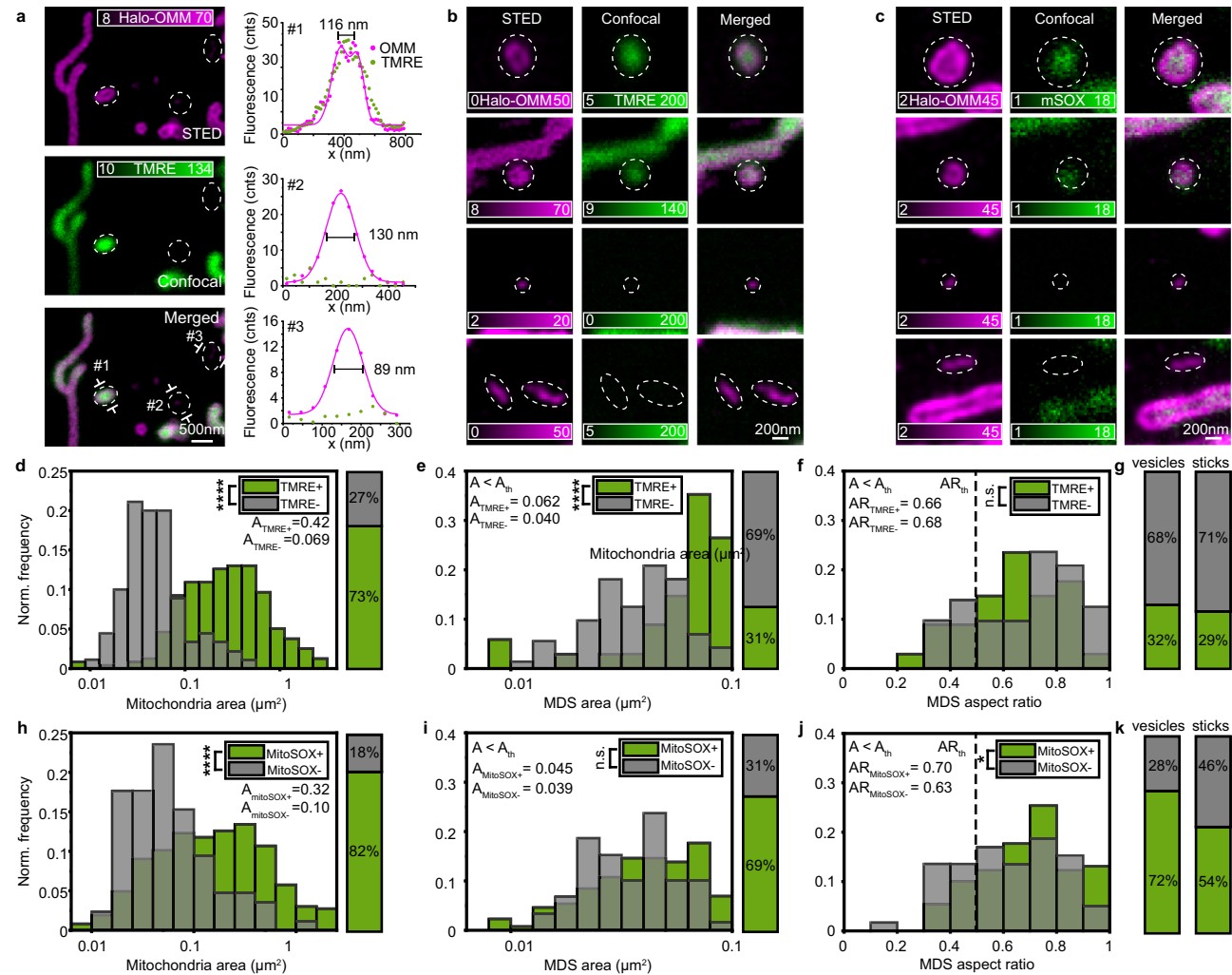

**Fig. 3 | MDSs are characterized by different functional states. a** Representative image of mitochondria stained for Halo-OMM (magenta) and tetramethyl rhodamine ethyl ester (TMRE), a membrane potential indicator (green), inside a neuronal cell. MDSs are highlighted with dotted lines. Line profiles measured across #1, #2, and #3 for TMRE (green dots) and OMM (pink dots for the data points and pink line for the Gaussian fit) show their sizes and intensities. The image is a representative example of the data plotted in (**d–g**). **b** Four examples of MDSs stained for Halo-OMM (magenta) and TMRE (green): (top) two vesicles with membrane potential, and (bottom) one vesicle and one stick without membrane potential. The images are representative examples of the data plotted in (**d–g**). **c** Four examples of MDSs stained for Halo-OMM (magenta) and reactive oxygen species (ROS) indicator (MitoSOX) (green); (top) two vesicles showing an accumulation of ROS, and (bottom) a vesicle and a stick without ROS. The images are representative examples of the data plotted in (**h–k**). **d** Distribution of the area of the entire mitochondria population, divided into mitochondria with (green) and without (gray) TMRE

signal. Data were collected from 9 DIV7 hippocampal neurons from four independent experiments. $N_{TMRE+} = 238$, $N_{TMRE-} = 90$. Kolmogorov–Smirnov (KS) test, ****$P < 0.0001$ ($P = 4.2 \times 10^{-29}$). **e, f** Distribution of the area and aspect ratio of MDSs, divided into MDSs with (green) and without (gray) TMRE signal. $N_{TMRE+} = 33$, $N_{TMRE-} = 72$. KS tests, ****$P < 0.0001$ ($P_{area} = 2.6 \times 10^{-7}$); $P_{aspect\ ratio} = 0.58$. **g** The ratio of vesicles and sticks with (green) and without (gray) TMRE signal. **h** Distribution and ratio of the area of the entire mitochondria population, divided into mitochondria with (green) and without (gray) MitoSOX signal. Data were collected from four DIV7 hippocampal neurons from two independent experiments. $N_{MitoSOX+} = 366$, $N_{MitoSOX-} = 85$. KS test: ****$P < 0.0001$ ($p = 4.2 \times 10^{-9}$). **i, j** Distribution of the area and aspect ratio of MDSs, divided into MDSs with (green) and without (gray) MitoSOX signal. $N_{MitoSOX+} = 130$, $N_{MitoSOX-} = 59$. KS tests: $P_{area} = 0.20$; *$P < 0.05$ ($P_{aspect\ ratio} = 0.019$). **k** The ratio of MDSs and sticks with (green) and without (gray) MitoSOX signal.

mitochondrial transcription factor A (TFAM) (Supplementary Fig. 3i), the most abundant protein inside mitochondrial nucleoids, where mitochondrial DNA (mtDNA) is located[33,34]. Using STED imaging, we identified single nucleoids of around 100 nm in size inside mitochondria (Supplementary Fig. 3j). We also spotted similar structures inside MDSs that were similar in size to those we had identified with membrane potential and ROS (Supplementary Fig. 3k). We therefore concluded that nucleoids, together with mitochondria functional state, size, and morphology, are pivotal for differentiating clearing MDSs from small mitochondria.

Altogether, we observed small vesicular and tubular MDSs enriched in ROS and without membrane potential, suggesting their role as MDVs involved in MQC. A second, larger population, with a larger area,

still below the resolution of a confocal microscope, has a membrane potential and can also contain mtDNA, thus resembling small mitochondria.

## MDS have different sizes and protein compositions
One of the major functions of mitochondria is to provide energy in the form of ATP, which is produced during oxidative phosphorylation. To further investigate MDSs composition and gain insight into their function, we examined the nanoscale distribution and abundance of oxidative phosphorylation complexes (OXPHOS) (Fig. 4). To visualize and quantify OXPHOS we used two different labeling strategies: for live-cell imaging we expressed full-length cytochrome C oxidase subunit 8A (COX8A) fused to a SNAP-tag (Supplementary Fig. 4a, b), while

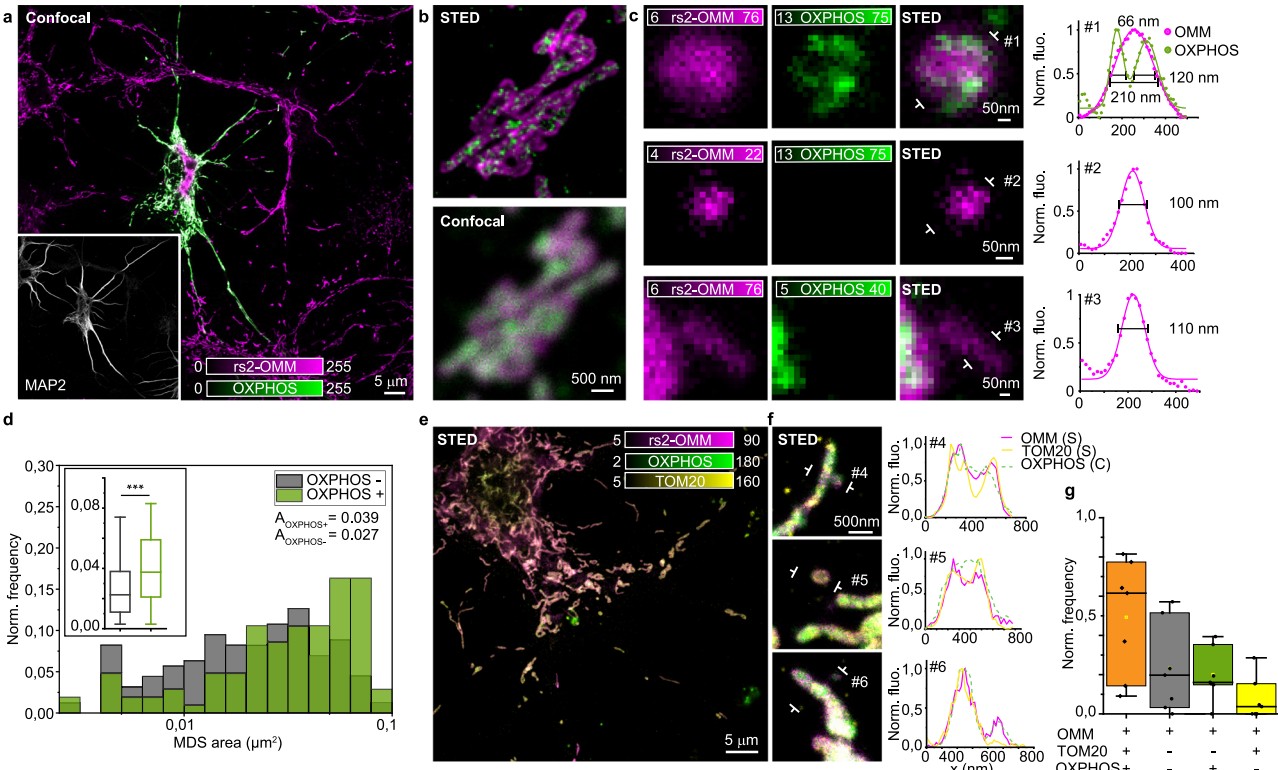

**Fig. 4 | MDSs have different sizes and protein compositions. a** Confocal image of a neuronal cell, representative of (**d**), expressing a rsEGFP2 fused to the OMM (rs2-OMM, magenta) and OXPHOS (green, immunostaining). The inset shows the dendrites (gray, MAP2 immunostaining). **b** Two-color STED and confocal comparison of mitochondria, representative of (**d**). The STED image, in contrast to the confocal, allows to discriminate between the OMM and the OXPHOS complexes organized in clusters along the inner mitochondrial membrane. **c** Three representative examples of MDSs and line profiles measured across #1, #2, and #3 in the images. In the bigger vesicle (FWHM = 210 nm), both OXPHOS and OMM are visible, while the smaller vesicle (FWHM = 100 nm) and the stick (FWHM = 110 nm) show the OMM only. **d** Distribution of MDS area divided into OXPHOS⁺ (green) and OXPHOS⁻ (gray). For the box plot, the center line represents the median, the box spans the interquartile range (IQR; 25th to 75th percentiles), and whiskers extend to 1.5× IQR. Data were collected from 15 DIV7–9 hippocampal neurons from three independent

experiments. $N_{tot}$ = 262 analyzed MDSs, $N_{OXPHOS−}$ = 158; $N_{OXPHOS+}$ = 104. KS tests, ***$P$ < 0.001 ($P$ = 3.8 × 10⁻⁴). **e** Two-color STED image of rs2-OMM (magenta) together with TOM20 (immunostaining, yellow), superimposed to a confocal image of OXPHOS complexes (immunostaining, green). **f** Three representative MDSs and the line profiles measured across #4, #5, and #6, showing two MDSs (#4 and #5) OMM⁺OXPHOS⁺TOM20⁺, and another one (#6) OMM⁺OXPHOS⁺TOM20⁻. **g** Box plot of the normalized frequency of MDS types with respect to the presence of TOM20 and OXPHOS complexes (OMM⁺TOM20⁺OXPHOS⁺: mean = 0.49, SD = 0.29, $N_{tot}$ = 134; OMM⁺ TOM20⁻ OXPHOS⁻: mean = 0.23, SD = 0.23, $N_{tot}$ = 61; OMM⁺ TOM20⁻ OXPHOS⁺: mean = 0.20, SD = 0.13, $N_{tot}$ = 54; OMM⁺TOM20⁺OXPHOS⁻: mean = 0.07, SD = 0.11, $N_{tot}$ = 8). For the box plot, the center line represents the median, the box spans the interquartile range (IQR; 25th to 75th percentiles), and whiskers extend to 1.5× IQR. Data were collected from 7 DIV7–9 hippocampal neurons, $N_{tot}$ = 257 MDSs.

for fixed cells we used an antibody cocktail targeting multiple OXPHOS subunits (subunits CI–V) (Fig. 4a). With STED, we identified the inner membrane ultrastructure, which was not distinguishable from the outer mitochondrial membrane by conventional microscopy (Fig. 4b). Live neuronal imaging of COX8A enabled us to visualize mitochondrial cristae as inner membrane invaginations of about 50 nm wide and 100 nm apart from each other (Supplementary Fig. 4a–c). Similar values were reported previously for HeLa cells[35]. Apart from labeling elongated tubular mitochondria, COX8A revealed a few puncta near mitochondrial tubules (Supplementary Fig. 4b inset), which could indicate double membrane MDSs. Time-lapse imaging of COX8A and OMM showed the different dynamics of inner and outer mitochondrial membranes, which are often asynchronously arranged: during the formation of a double membrane mitochondrial protrusion, the inner membrane detached first, while the outer membrane remained connected to the mitochondrion (Supplementary Movie 4). OXPHOS-cocktail stained mitochondria as small clusters of 60–100 nm in size, next to the OMM (Fig. 4b). The OXPHOS composition of the MDSs was highly variable between cells (Supplementary Fig. 4d): only some MDSs contained respiratory complexes, while most small vesicles and sticks lacked OXPHOS (Fig. 4c, d). We additionally found that areas of OXPHOS⁺ MDSs were significantly larger than those of

OXPHOS⁻ MDSs (mean area: $A_{OX+}$ = 0.048 μm², $A_{OX−}$ = 0.027 μm²; mode area: $A_{OX+}$ = 0.028 μm², $A_{OX−}$ = 0.021 μm²) (Fig. 4d). The same trend was true for length and width, while their ARs were not significantly different (Supplementary Fig. 4e–g). Our results suggest that a small group of larger MDSs likely represent functionally active mitochondria, with membrane potential, mtDNA, and OXPHOS. However, smaller MDSs (width <150 nm) usually lack membrane potential, accumulate ROS, and may or may not contain OXPHOS. We therefore added an additional staining for the outer mitochondrial membrane, the import protein TOM20, which was also shown to localize exclusively in clearing vesicles directed to lysosomes[11]. By immunolabelling TOM20, we revealed mitochondria, as expected, but also MDSs, which were OXPHOS⁺. Moreover, we identified a small population of TOM20⁺ and OXPHOS⁻ (3%), likely representing MDVs involved in the MQC directed to lysosomes (Fig. 4e–g and Supplementary Fig. 4h). These vesicles were on average smaller in size than the others (Supplementary Fig. 4i). We found that 24% of MDSs were OXPHOS⁺ but TOM20⁻. We hypothesized that this population could be involved in a different, yet unknown function, since they contain OXPHOS complexes, but do not target lysosomes. In support of this, Hazan et al. recently showed that isolated mitochondria from *Saccharomyces cerevisiae* release MDVs carrying a functional ATP synthase complex and proposed that

these vesicles fuse with naïve mitochondria and regenerate ATP-deficient mitochondria[36].

Overall, we found that double membrane MDSs with OXPHOS are typically larger than those without and can be involved in organelle-to-organelle communication. Within this population, a subset containing TOM20 is significantly larger and likely represents small, intact mitochondria. Vesicles containing TOM20, but not OXPHOS, are the smallest in size and potentially represent MDVs involved in MQC.

### Perturbation of mitochondria respiration affects MDS

We characterized the size and morphology of MDSs, and further analyzed their functionality, together with a thorough analysis of OXPHOS and mtDNA content to identify MDVs within the MDS population. To gain insights into their regulation, we perturbed mitochondria respiration by treating neurons with antimycin A (AA), a drug that inhibits mitochondrial electron transport and leads to ROS production

(Fig. 5 and Supplementary Fig. 5). Firstly, neurons were treated with low doses of AA (5 nM) for a prolonged time (6 h) to mimic the mild and reversible mitochondrial stress characteristics of the initial stages of some neurodegenerative diseases[22]. We refer to this treatment as mild stress. After applying mild stress, we observed mitochondria with STED, counted them, and analyzed their morphologies (Fig. 5a, b). With Mitography, we observed reduced area and length values in treated cells compared to controls. The AR, however, showed increased values (Supplementary Fig. 5a–h). These morphological changes were even more pronounced when we induced a strong stress (40 μM AA, 1 h) (Supplementary Fig. 5i–l). The strong stress, however, likely induced mitophagy, since we observed mitochondria fragmentation, and the AR shifted towards higher values (Supplementary Fig. 5m). Mild stress, in contrast, did not alter mitochondrial number or area, but induced a significant increase in MDS, further highlighting the existence of MDVs involved in stress-induced MQC (Fig. 5c–e and

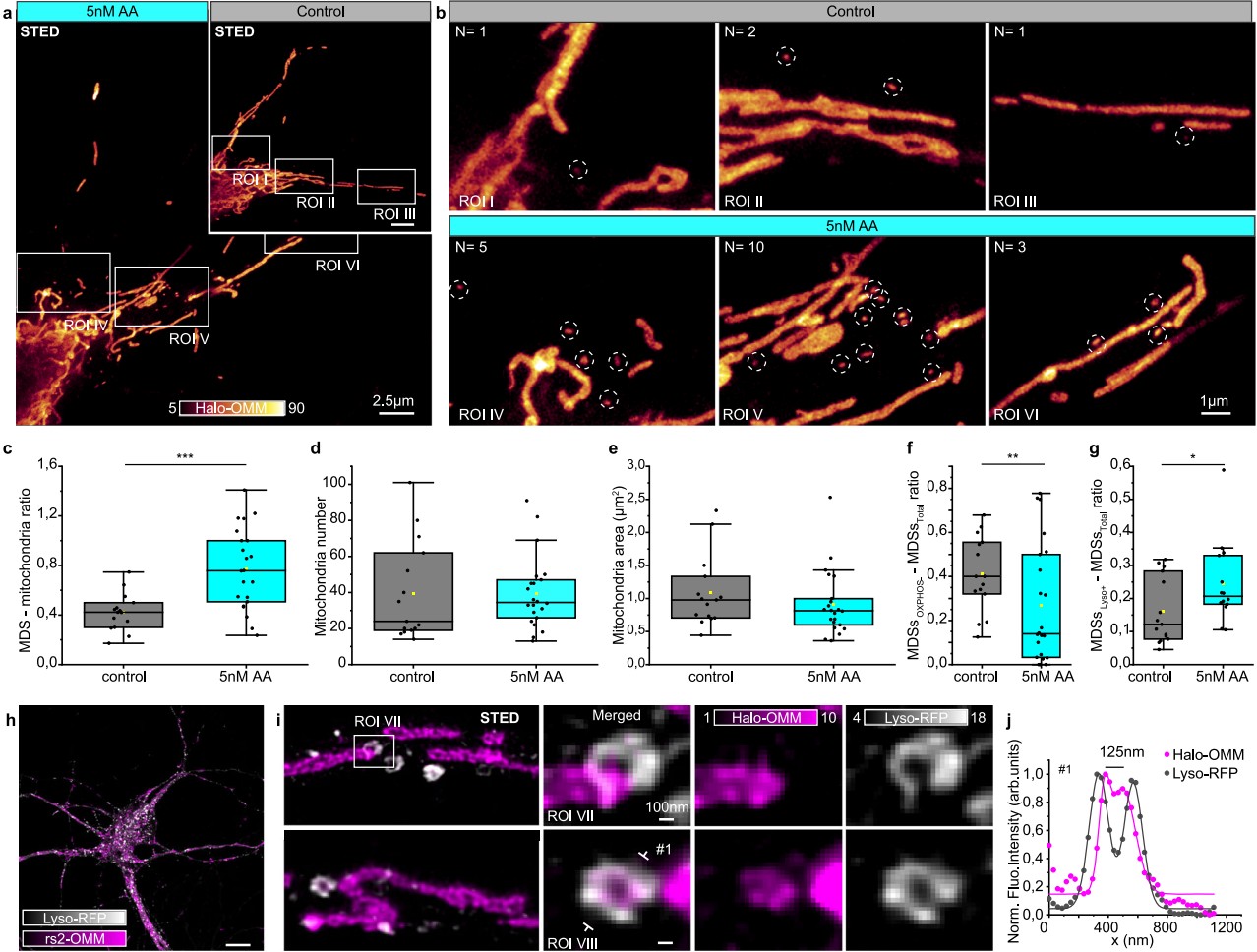

**Fig. 5 | Perturbation of mitochondria respiration affects MDSs. a** STED images (representative of (**c**–**f**)) of neuronal cells expressing a Halo-tag fused to the OMM (Halo-OMM), grown with antimycin A (AA) or without (control, inset). **b** Three representative ROIs from (**a**), showing the increased number of MDSs (highlighted with dotted circles) in the treated cell with respect to the control. **c** Box plots of MDS/mitochondria ratio, showing a significant increase of MDSs upon AA treatment. $N_{control} = 15$ neurons, $N_{AA} = 22$ neurons, from three independent experiments. Each data point represents one cell. Student's $t$-tests, ***$P < 0.001$ ($P = 4.35 \times 10^{-4}$). **d**, **e** Box plots of the mitochondrial number and mitochondrial area per cell for control and AA-treated cells. $N_{control} = 15$ cells, $N_{AA} = 22$ cells, from three independent experiments. Each data point represents one cell. Kolmogorov–Smirnov test, $p_{mito\#} = 0.29$; $p_{Area} = 0.39$. **f** Box plot of the ratio between the MDSs$_{OXPHOS+}$ and the total number of MDSs for control and AA-treated cells. $N_{control} = 15$ cells, $N_{AA} = 22$

cells, from three independent experiments. Each data point represents one cell. KS test, **$P < 0.01$ ($P = 0.009$). **g** Box plot of the ratio between the MDSs colocalizing with lysosomes (MDS Lyso+) and the total number of MDSs for control and AA-treated samples. $N_{control} = 15$ cells, $N_{AA} = 14$ cells, from three independent experiments. Each data point represents one cell. KS test, *$P < 0.05$ ($P = 0.02$). **h** Confocal image of a neuron, representative of (**g**), labeled for the OMM (rs2-OMM) and Lamp1 (Lyso-RFP). **i** STED images of Lamp1 (Lyso-RFP) and OMM (rs2-OMM), a representative example of the data plotted in (**g**), zoom-in showing lysosomal structures delimiting the tip of a mitochondrial tubule (ROI VII) and an MDS (ROI VIII). **j** Normalized line profiles from ROI VIII #1 (dots are data points; lines are Gaussian fits). Box plots in (**c**–**g**): the center line represents the median; the box spans the interquartile range (IQR; 25th to 75th percentiles), and whiskers extend to 1.5× IQR.

Supplementary Fig. 5n–p). We further investigated the composition of these MDSs and observed that AA treatment reduced the number of those enriched in OXPHOS (Fig. 5f). This result is in line with our previous observation that OXPHOS is excluded from smaller vesicles, which instead showed ROS accumulation and lacked TMRE staining. Altogether, these results agree with the proposed role of MDVs in MQC. To further confirm these findings, we performed the mild AA treatment and examined interactions with lysosomes, the waste disposal system within the cell (Fig. 5g–j). We labeled them via exogenous tagging of the lysosomal-associated membrane protein 1 (Lamp1) with the Red Fluorescent Protein (RFP), followed by immunostaining against the fluorescent tag and quantifying the degree of interaction with MDSs in controls and treated samples (5 nM AA, 6 h). Upon mitochondrial stress, the number of MDSs interacting with lysosomes increased significantly (Fig. 5g). Moreover, STED microscopy allowed us to observe the engulfment of mitochondrial protrusions and MDSs by lysosomal structures (Fig. 5i, j).

Overall, these results suggest that neurons are likely to adapt to metabolic stress by releasing MDSs in the form of MDVs targeting lysosomes, to mitigate damaged mitochondrial components and prevent mitophagy.

## Diverse MDSs connect with peroxisomes in neurons

We saw that the MDS population is highly heterogeneous and clearing MDVs target lysosomes to support mitochondria during stress conditions. Another group of MDVs was previously reported to shuttle cargoes to a subpopulation of peroxisomes[18] and to be involved in their biogenesis[37,38]. Peroxisomes are membrane-bound oxidative organelles that share important cellular functions with mitochondria, such as fatty acid oxidation and peroxide reduction[39,40]. However, interactions between mitochondria and peroxisomes in neurons have not been studied in detail. To investigate the relationship between MDSs and peroxisomes, we labeled the peroxisome biogenesis protein PEX14 together with the OMM (Fig. 6). We identified many PEX14 puncta located close to mitochondria (Fig. 6a). STED microscopy showed the distribution of PEX14 and its organization with respect to the mitochondrial outer membrane (Fig. 6b). PEX14 was also found in mitochondrial protrusions, vesicles, and sticks (Fig. 6c). Unexpectedly, most (86%) of the elongated sticks colocalized with PEX14. Overall, 76% of MDSs, including vesicles and sticks, colocalized with PEX14. PEX14$^+$ MDSs were on average significantly larger than PEX14$^-$ (mean area: $A_{PEX+} = 0.035\ \mu m^2$, $A_{PEX-} = 0.028\ \mu m^2$) (Fig. 6d and Supplementary Fig. 6a–c). Most of the MDSs colocalizing with PEX14

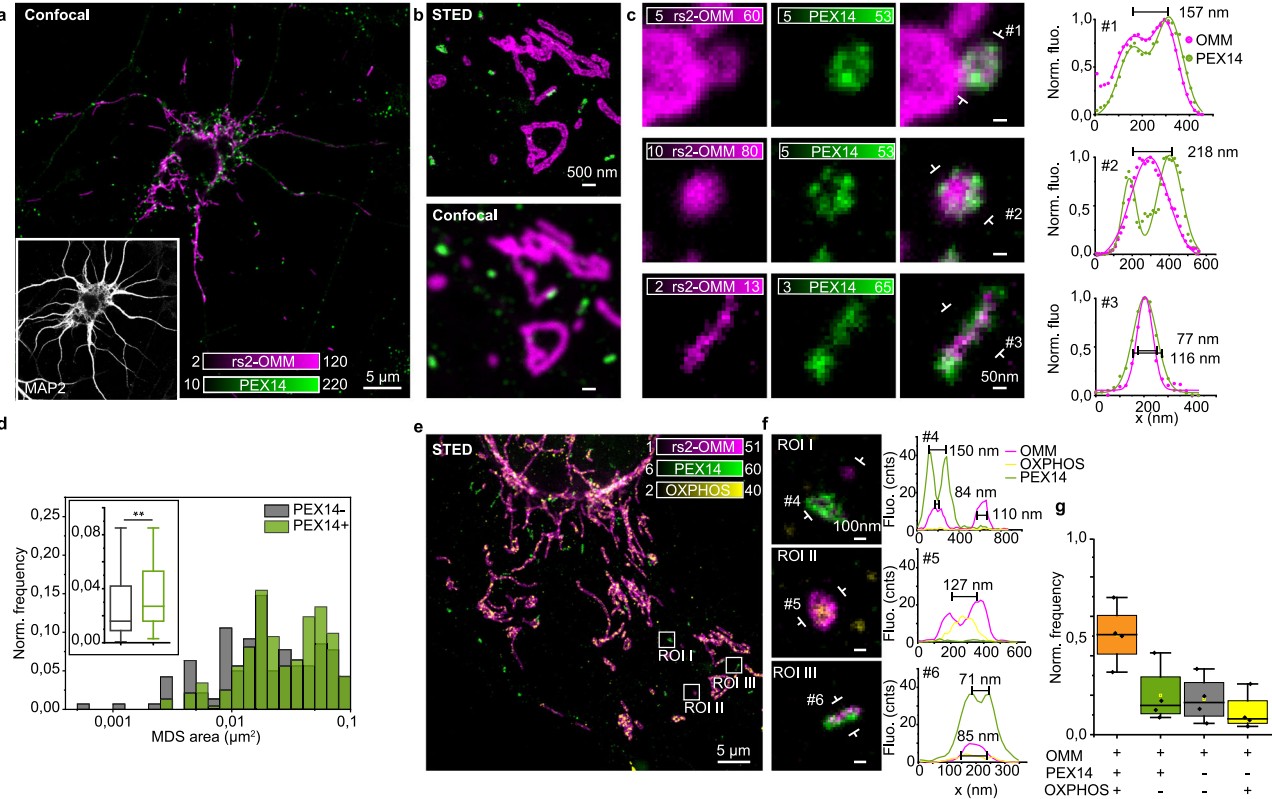

**Fig. 6 | Diverse MDSs connect with peroxisomes in neurons. a** Confocal image, representative of (**d**), of a neuronal cell expressing the rsEGFP2 fused to the OMM (rs2-OMM, magenta) and immunostained for the peroxisomal protein PEX14 (green). The inset shows the dendrites, immunostained for MAP2 (gray). **b** Two-color STED and confocal comparison of multiple mitochondria and peroxisomes, representative of (**d**). STED image, in contrast to the confocal comparison, allows to appreciate the outer mitochondrial membrane and the peroxisomal membrane. **c** Three representative examples of MDSs and line profiles measured across #1, #2, and #3 in the images. (Top) MDS from the tip of a mitochondrion, with OMM and PEX14 signals colocalizing in the vesicle membrane, indicating the possible biogenesis of a new pre-peroxisome; (middle) MDS where OMM is localized inside PEX14; (bottom) MDS with OMM and PEX14 signals, indicating another example of possible mitochondria-peroxisome exchange mechanisms. **d** Distribution of MDS area divided into PEX$^+$ (green) and PEX$^-$ (gray). Data were collected from six DIV7–9

hippocampal neurons from two independent experiments. MDS$_{OXPHOS-}$ = 142; MDS$_{OXPHOS+}$ = 233. Kolmogorov–Smirnov test, **$P$ < 0.01. **e** Two-color STED image of rs2-OMM (magenta) and the peroxisomal protein PEX14 (immunostaining, green), superimposed to a confocal image of the OXPHOS complex (immunostaining, yellow). The image is a representative example of the data plotted in (**g**). **f** Three examples of MDSs and line profiles at the indicated positions from (**e**), showing the absence of OXPHOS in PEX$^+$ structures. **g** Box plot of the normalized frequency of MDS types with respect to the presence of PEX14 and OXPHOS (OMM$^+$PEX14$^+$OXPHOS$^+$: mean = 0.51, SD = 0.15, $N_{tot}$ = 71; OMM$^+$PEX14$^+$OXPHOS$^-$: mean = 0.2, SD = 0.15, $N_{tot}$ = 31; OMM$^+$PEX14$^-$OXPHOS$^-$: mean = 0.18, SD = 0.12, $N_{tot}$ = 29; OMM$^+$PEX14$^-$OXPHOS$^+$: mean = 0.11, SD = 0.09, $N_{tot}$ = 16). Data were collected from four neurons, $N_{tot}$ = 147 MDSs. For the box plots in (**d**, **g**), the center line represents the median, the box spans the interquartile range (IQR; 25th to 75th percentiles), and whiskers extend to 1.5× IQR.

also contained OXPHOS (Fig. 6e–g and Supplementary Fig. 6d), which might represent double-membrane vesicles targeted to peroxisomes. However, there was also a fraction of MDSs (20% of the population) that colocalized with PEX14, but not with OXPHOS. Upon low doses of AA, the ratio of PEX+ MDSs remained constant, suggesting that these vesicles are not affected by mitochondrial stress and therefore are not involved in MQC (Supplementary Fig. 6e–g).

Because of the shared functions in lipid metabolism and the involvement of MDVs in peroxisome biogenesis[38], we investigated the lipid composition of MDSs (Supplementary Fig. 7). By staining both the OMM and lipids (Nile Red), we demonstrated the coexistence of peptides and lipids in MDSs (Supplementary Fig. 7a, b). We further used 10-N-nonyl acridine orange (NAO) to stain cardiolipin, which is almost exclusively found in the mitochondrial inner membrane[41] (Supplementary Fig. 7c). As expected, NAO accumulated inside mitochondria but was not found in small vesicles and sticks (Supplementary Fig. 7d, e). The absence of cardiolipin supports our observation that smaller vesicles and sticks lack inner mitochondrial membrane components such as OXPHOS.

Altogether, these results indicate the existence of at least two MDS types interacting with peroxisomes, differing in composition and possibly function.

## MDSs participate in forming pre-peroxisomes at mitochondria

To test whether this population of peroxisome-connected MDSs could be involved in peroxisome de novo biogenesis, we first performed a pulse-chase experiment, based on enzymatic labeling, and characterized the nature and lifetime of this population of small vesicles showing both mitochondrial and peroxisomal content. Neurons transfected with OMP25-SNAP were pulsed for 1 h with a non-fluorescent ligand to cover all available SNAP-tag expressed at this time point ($t = 0$) and chased for 24 h. After 24 h ($t = 24$ h), cells were pulsed with the SNAP ligand SiR-647-BG to label newly synthesized

membranes. After fixation, neurons were stained for PEX14 (Fig. 7a). An abundant population of labeled PEX14 puncta (38%) colocalized with newly produced MDSs (Fig. 7b). In these newly formed small vesicles, no TMRE signal was detectable, suggesting altered membrane potential (Fig. 7c–e). Overall, our results demonstrate the presence of a distinct and prominent population of MDSs (70–150 nm), likely not involved in MQC, but instead being MDVs contributing to peroxisome biogenesis, as supported by the discovery of newly translated protein content inside these vesicles.

Another key protein involved in the de novo peroxisome biogenesis is the peroxisomal biogenesis factor 3 (PEX3), a core component of the machinery required for the assembly of peroxisome membrane vesicles prior to the import of matrix proteins. PEX3 is also known to be involved in the process of pre-peroxisome formation via MDVs[38]. To further test the MDS's connection with de novo biogenesis, we silenced PEX3 expression and examined the consequences in neuronal cells (Supplementary Fig. 8 and Supplementary Note 2). Three commercially available siRNAs targeting distinct regions of the PEX3 transcript were used. PEX3 protein levels were assessed 24 and 48 h post-transfection via immunostaining. Quantification revealed a significant reduction in the fluorescence intensity of PEX3 punctae ($p = 7.68462 \times 10^{-4}$) and a decrease in their linear number density ($p = 0.01075$) relative to non-transfected controls (Supplementary Fig. 8a–d). Similarly, mature peroxisomes labeled with PMP70 showed a reduced density ($p = 0.00903$), demonstrating that transient PEX3 knockdown effectively impairs de novo peroxisome formation (Supplementary Fig. 8e–g).

We then employed STED to investigate de novo peroxisome biogenesis and examined the distribution of vesicular protrusions with respect to the mitochondrial membrane to determine whether there was a preferential site of vesicle formation (Fig. 8). Only those attached to mitochondria and labeled with OMM were considered. Most of the identified protrusions were found at the tip of tubular mitochondria

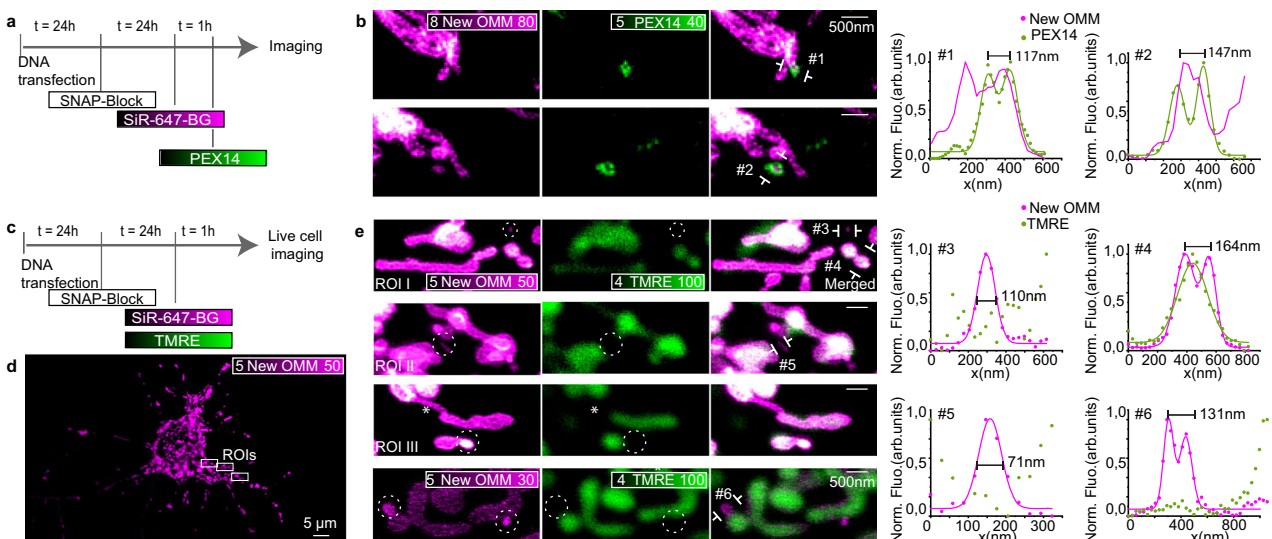

**Fig. 7 | MDSs participate in forming pre-peroxisomes at mitochondria.**
**a** Timeline scheme of SNAP-tag protein expression system to visualize new outer mitochondrial membranes (New OMM) together with the peroxisomal membrane protein PEX14. **b** Two representative examples of MDSs stained through the C-terminal localization peptide of the mitochondrial outer membrane protein OMP25 fused to the SNAP-tag (New OMM, magenta) and PEX14 (immunostaining, green), and line profiles measured across lines #1 and #2 are indicated in the images. The experiment was repeated twice, for a total of 8 neurons. **c** Timeline

scheme of SNAP-tag protein expression system to visualize New OMM together with the membrane-potential-sensitive dye TMRE. **d** Representative confocal image of a neuronal cell where new SNAP-OMM (New OMM) is labeled. **e** Four representative examples of MDSs stained for new SNAP-OMM (New OMM, magenta) and TMRE dye (green), and line profiles measured across lines #3, #4, #5, and #6 indicated in the images. Multiple MDSs with new membranes (dashed outlines) do not have a membrane potential. The experiment was repeated twice, for a total of 15 neurons.

(Fig. 8a, c), in accordance with the recently proposed model for TOM20+ vesicle formation[42]. However, we also saw membrane protrusions budding out from the lateral side of the mitochondrial tubule (Fig. 8b, c). We then used multicolor STED to localize markers for pre- and mature peroxisomes and examined their distribution within the protrusions. De novo peroxisome biogenesis has recently been shown to require both ER- and mitochondria-targeted proteins. In fact, integral peroxisomal membrane proteins such as PEX3 and PEX14 localize to mitochondria in cells lacking peroxisomes and are released in the form of vesicles, which, upon fusion with PEX16+ ER-derived vesicles,

lead to the generation of pre-peroxisomes[38]. However, the precise location of vesicle fusion and the mechanism of peroxisome biogenesis are still unknown. Here, pre-peroxisomes were identified by the double labeling of PEX14 and PEX16 (Fig. 8d), while mature peroxisomes were labeled by the membrane protein PMP70 (Fig. 8e). Most of the vesicular protrusions were stained by two markers, either PEX14+PEX16+ or PEX16+PMP70+, whereas single marker structures were less common (Fig. 8f, g). When we measured the sizes of all protrusions, those labeled by PEX14+PEX16+ were clearly distinct from those labeled by PEX16+PMP70+: PEX14+16+ vesicles were significantly

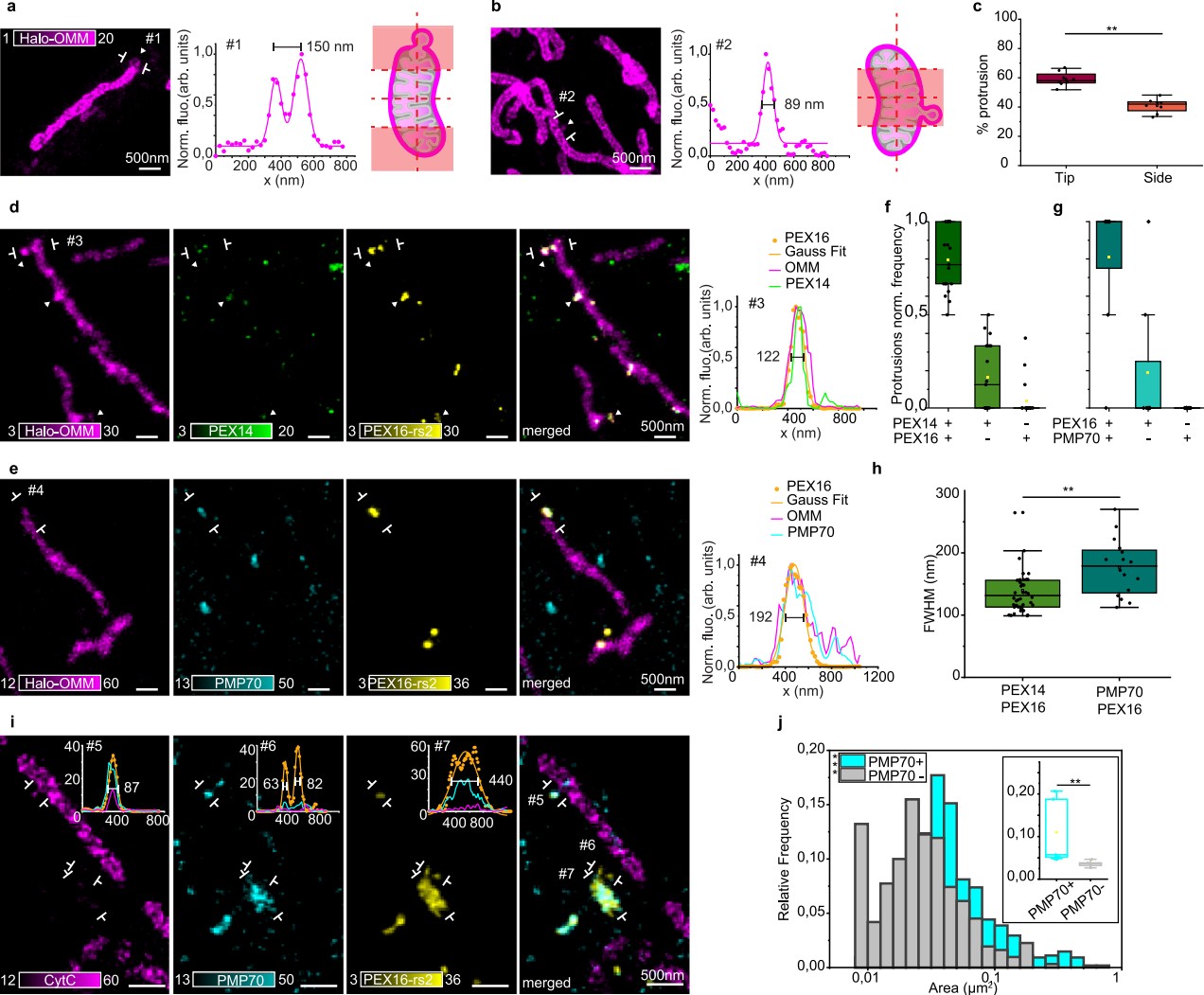

**Fig. 8 | MDSs contribute to the de novo biogenesis of peroxisomes.**
**a** Representative example (from **c**) of a mitochondrial protrusion at the tip of a mitochondrion, relative line profile at the indicated signs, and a relative schematic to show the area considered as "tip". **b** Representative example (from **c**) of a mitochondrial protrusion at the side of a mitochondrion, relative line profile at the indicated signs, and relative schematic to show the area considered as "side".
**c** Percentages of protrusions found at the tip or at the side. $N_{tot}$ = 800, $N_{tip}$ = 473, $N_{side}$ = 327, more than three independent experiments. Each point represents one experiment. Wilcoxon Signed Ranks Test, **$P$ < 0.01 ($P$ = 0.008). **d** Representative example (from **f**, **h**) showing mitochondria (Halo-OMM, magenta), PEX14 (immunostaining, green), and PEX16-rsEGFP2 (exogenous PEX16-rs2, yellow). The indicated line profile shows the colocalization of the three at the mitochondrial tip.
**e** Representative example (from **g**, **h**) showing mitochondria (Halo-OMM, magenta), PEX16-rsEGFP2 (exogenous PEX16-rs2, yellow), and PMP70 (immunostaining, cyan). The indicated line profile shows the colocalization of the three at the mitochondrial tip. **f** Box plot of normalized mitochondrial protrusion frequency

across PEX16+PEX14−, PEX14+PEX16−, and PEX14+PEX16+ groups. PEX14−PEX16− are excluded (roughly 50% of total). $N_{protrusions}$ = 203 protrusions, 20 cells, 3 different experiments. **g** Box plot of normalized mitochondrial protrusion frequency across PMP70+PEX16−, PEX16+PMP70−, and PMP70+PEX16+groups. PMP70−PEX16− are excluded (roughly 50% of the total). $N_{protrusions}$ = 127, 9 cells, 2 different experiments. **h** Box plot of the FWHM of PEX14+/PEX16+ and PMP70−/PEX16+ mitochondria protrusions. Each data point is one protrusion. PEX14+/PEX16+ $N_{protrusions}$ = 42, three independent experiments; PEX16+/PMP70+ $N_{protrusions}$ = 16, two independent experiments. KS test, **$P$ < 0.01. **i** Representative example of the data in (**j**) of PEX16-rsEGFP2+ (exogenous PEX16-rs2, yellow) vesicles colocalizing (5) or not (#3,#4) with mature peroxisomes (PMP70 immunostaining, cyan) and mitochondria (CytC immunostaining, magenta) and relative line profiles. **j** Histogram and box plot of PEX16+ MDS area divided into PMP70+ and PMP70−. For the box plot, the center line represents the median, the box spans the interquartile range (IQR; 25th to 75th percentiles), and whiskers extend to 1.5× IQR. Histogram: KS test, $p = 2.66 \times 10^{-18}$; box plot: KS test, **$P$ < 0.01.

smaller (mean size: 140 nm) than PEX16$^+$PMP70$^+$ vesicles (mean size: 177 nm) (Fig. 8h). The size difference likely reflects the "age" of the peroxisomal structures identified here: immature pre-peroxisomes (PEX14$^+$16$^+$) are smaller than the mature ones PEX16$^+$PMP70$^+$. To further confirm this observation, we measured the sizes of the overall peroxisome population, identified in this case by the double staining of PEX16$^+$PMP70$^+$ (Fig. 8i). Peroxisomes showed a wide range of sizes and shapes, with PMP70$^+$ structures being bigger than PMP70$^-$, likely to indicate the presence of mature peroxisomes (PEX16$^+$PMP70$^+$) and small vesicles, or possible peroxisome precursors, shuttling between mitochondria and peroxisomes (PEX16$^+$PMP70$^-$) (Fig. 8j). These results point to the hypothesis that pre-peroxisome vesicle generation happens at mitochondria sites, via the fusion of PEX14$^+$ and PEX16$^+$ vesicles.

PEX14$^+$16$^+$ vesicles, still attached to or in contact with the mitochondrial membrane, may represent the sites where mitochondrial and ER proteins fuse together to give rise to new peroxisomes. How does this process happen? Which are the players involved? To answer these questions, we first performed two-color time-lapse STED imaging and monitored mitochondrial vesicle formation in the presence of either ER or peroxisomal markers (Fig. 9). We observed mitochondrial vesicles released from both the lateral side and the tip of mitochondrial tubules and in both cases ER tubules are present at the sites of vesicle release (Fig. 9a, b and Supplementary Movies 5 and 6). These time-lapses resemble the process happening during mitochondrial fission, where ER tubules wrap around the site of constriction to favor fission[31], suggesting a similar involvement of ER in MDS formation. Moreover, the proximity of ER structures at sites of mitochondrial protrusion could facilitate the encounter between mitochondria and ER-derived vesicles, a condition required for the de novo peroxisome biogenesis process. To specifically identify pre-peroxisomal protrusion, we therefore labeled either PEX16 (Fig. 9c) or PEX14 (Fig. 9d) and confirmed the presence of the ER, marking the site between the elongated tubule and the protrusion constriction. These results overall showed that most of the analyzed mitochondrial protrusions that are colocalizing with either PEX16 or PEX14 also have an ER signal nearby (Fig. 9e, f). Time-lapse confocal and STED imaging revealed mitochondrial protrusions colocalizing with PEX14-GFP. These protrusions detached from mitochondrial tubules to form MDSs associated with peroxisomal markers (Supplementary Fig. 9a, b). Another fundamental player in mitochondrial fission is the dynamin-related protein 1 (DRP1). Therefore, we looked at the distribution of DRP1 protein in relation to mitochondrial protrusions. We found DRP1 at most of the identified protrusions, colocalizing with ER or PEX14 (Fig. 9g, h). Next, to perturb DRP1 function, we expressed DRP1K38A mutant (Supplementary Note 2), which harbors a dominant-negative mutation in the guanosine triphosphate (GTP)-binding pocket, thereby inhibiting GTP binding and hydrolysis[43]. Previous studies have shown that this mutant exhibits reduced efficiency in mediating mitochondrial constriction and division[44] Consistent with these reports, our results show that DRP1K38A overexpression induces the formation of elongated mitochondrial tubules and promotes an overall elongation of the mitochondrial network relative to cells expressing DRP1 wild type (Supplementary Fig. 9c–e). Analysis of the spatial relationship between mitochondria and pre-peroxisomal markers, such as PEX3 and PEX14, revealed a significant reduction in the number of pre-peroxisomal punctae contacting mitochondria in neuronal cells expressing DRP1K38A compared to control cells (Supplementary Fig. 9f–h), as measured by the linear number density of contact sites per mitochondrial length. In contrast, no differences were observed in the contact sites between mitochondria and mature peroxisomes labeled with PMP70 (Supplementary Fig. 9i). Collectively, these findings indicate that disruption of DRP1 function compromises mitochondrial dynamics and selectively impairs the association between mitochondria and pre-peroxisomal structures, without affecting mature

peroxisome interactions. They further suggest a potential role for DRP1 in the regulation of de novo pre-peroxisome biogenesis mediated by MDSs. Altogether, these results show that ER and DRP1 mark sites of mitochondrial vesicle protrusions, which may detach from the mitochondrial tubule to form pre-peroxisomes (Fig. 9i).

In conclusion, the findings from both pulse-chase experiments and STED imaging provide compelling evidence for a distinct and significant population of small MDSs contributing to peroxisome biogenesis, shedding light on the mechanism of de novo peroxisome biogenesis, and emphasizing the role of the mitochondria-ER connection and fission machinery in the process of vesicle fusion and pre-peroxisome formation.

## Discussion

In this study, we developed several labeling strategies to combine functional imaging and super-resolution microscopy to elucidate the structural and functional relationship of MDVs in living neuronal cells. Neurons possess distinct compartments with specialized architectures and functions, potentially leading to differences in MDVs between axons and dendrites. This study aimed to systematically analyze MDVs in neuronal cells, from their morphology and size to their functions, using fluorescent tagging. Although conventional microscopy is suitable to visualize relatively large mitochondrial organelles, super-resolution microscopy is necessary to identify fine structures, such as outer mitochondrial membranes[45,46] and mitochondrial-derived vesicular and tubular organelles, often located near larger mitochondria[28,47]. Here, we acquired and analyzed STED images with 30–50 nm spatial resolution to uncover the ultrastructure of mitochondria and MDVs in small and densely packed neuronal subcompartments in live cells. Applying a custom-written automated analysis, we systematically measured and characterized the size and morphology of mitochondria and differentiated smaller structures ($A < 0.086\ \mu m^2$), which we defined as MDSs. At this stage, this category may include MDVs, mitochondrial fragments, and small functional mitochondria. We therefore combined well-established MDV markers with precise measurements of size and morphology to characterize the different subsets of MDSs present in neuronal cells, with specific attention to MDVs. MDSs have both round and stick-like shapes, are delimited by the outer mitochondrial membrane, and can also contain the inner membrane and matrix components. Both MDS populations were observed throughout the neuron, but vesicles were relatively more abundant in dendrites, while sticks were more frequently found in the soma. Time-lapse STED imaging revealed the mitochondrial origin of MDSs, allowing us to follow membrane protrusion and vesicle release, and to gain insight into the mechanism of vesicle scission. Indeed, we observed ER tubules and DRP1 accumulation at sites of vesicle formation, suggesting similarities with the mechanism of mitochondria fission[31]. These data are supported by a recent study showing the presence of DRP1 at sites of MDV release[42] and imply the involvement of fission-machinery components in the process of vesicle release.

To define the nature of these structures, we combined functional readouts with ultrastructural observations, revealing distinct populations of MDSs, including different types of MDVs. We found a subset of MDSs (mean area = $0.04\ \mu m^2$) that do not have membrane potential, but do accumulate ROS, suggesting their possible role as MDVs involved in MQC. Similar vesicles have been reported in other cell types, such as cardiac myoblasts[48]. Upon mild and prolonged AA treatment, a protocol previously used to simulate the chronic mitochondrial stress happening in neurodegenerative diseases[22], neurons showed an increased number of MDSs but an unchanged mitochondria population. The increased number of MDSs colocalizing with TOM20 and LAMP1, but not with other markers such as OXPHOS, supports the role of these smallest structures as MDVs involved in MQC. These vesicles operate in basal conditions but additionally

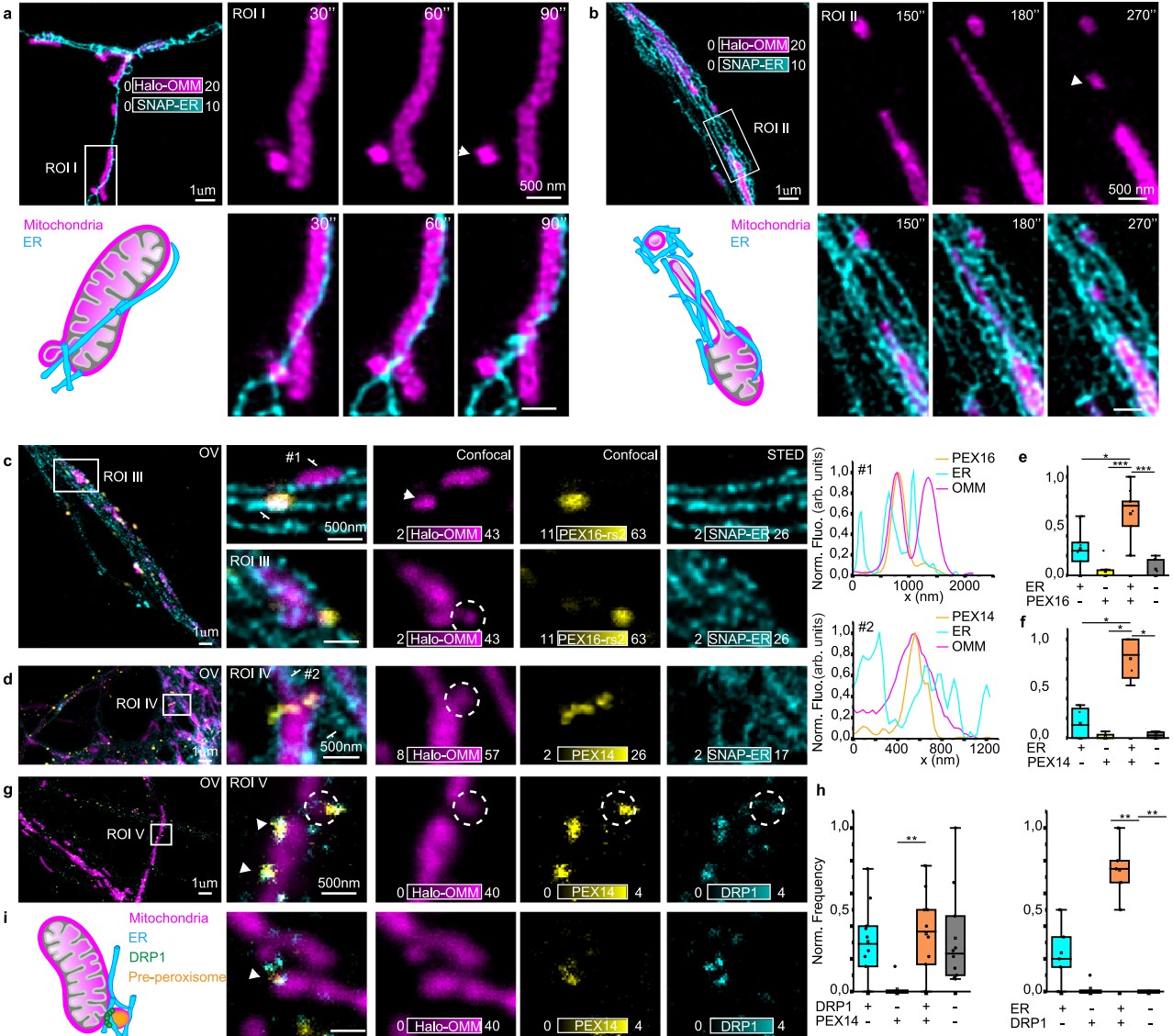

**Fig. 9 | ER and DRP1 mark the site of pre-peroxisome formation. a** Two-color STED image of ER (SNAP-ER, cyan) and mitochondria (Halo-OMM, magenta), representative of more than three independent experiments. ROI I shows detachment of a lateral protrusion at an ER contact-site (see Supplementary Movie 5). The schematic illustrates ER-mitochondria interactions during mitochondrial-vesicle release. **b** Two-color live STED image of ER (SNAP-ER, cyan) and mitochondria (Halo-OMM, magenta), representative of more than three independent experiments. ROI II shows detachment of a lateral protrusion at an ER contact-site (see Supplementary Movie 6). The schematic illustrates ER-mitochondria interactions during mitochondrial-vesicle release. **c** Three-color STED image of mitochondria, PEX16 and ER, representative of (**e**), showing MDS and mitochondrial protrusion with PEX16 enrichment, with ER tubules marking the pre-peroxisome formation site, confirmed by the line profile at position #1. **d** Three-color STED image of mitochondria, PEX14 and ER, representative of (**f**), showing a mitochondrial protrusion with PEX14 enrichment, with ER surrounding the protrusion, confirmed by the line profile at position #2. **e** Box plot of

normalized mitochondrial protrusion frequency relative to proximity to ER and/ or PEX16. Each data point represents the average number of protrusions per neuron; $N_{neurons} = 9$, $N_{protrusions} = 102$. KS tests; *$P < 0.05$ ($P = 0.03$), ***$P < 0.001$ ($P = 7.4 \times 10^{-4}$; $P = 7.4 \times 10^{-4}$). **f** Box plot of normalized mitochondrial protrusion frequency relative to proximity to ER and/or PEX14. Each data point represents the average number of protrusions per neuron; $N_{neurons} = 4$, $N_{protrusions} = 69$. KS tests; *$P < 0.05$. **g** Representative three-color STED image, relative to (**h**) of mitochondria (Halo-OMM, magenta), PEX14 (immunostaining, yellow), and DRP1 (immunostaining, cyan) showing DRP1 at PEX14+ mitochondrial-protrusion sites. **h** Box plots of normalized mitochondrial protrusion frequency relative to proximity to DRP1 and/or PEX14 ($N_{neurons} = 10$, $N_{protrusions} = 112$) and DRP1 and/or ER ($N_{neurons} = 5$, $N_{protrusions} = 37$). Each data point represents the average number of protrusions per neuron. KS tests; **$P < 0.01$ ($P = 0.002$; $P = 0.008$). **i** Sketch of mitochondria, ER, and DRP1 at pre-peroxisome formation site. For the box plots in (**e**, **f**, **h**), the center line represents the median, the box spans the interquartile range (IQR; 25th to 75th percentiles), and whiskers extend to 1.5× IQR.

---

serve as an initial response during early stages of mitochondrial stress or when the damage is localized, preventing mitophagy and the consequent degradation of the full mitochondria[32,49]. A distinct subset of larger vesicles (mean area = 0.06 µm²) instead has membrane potential, while containing OXPHOS complexes and the mitochondrial transcription factor TFAM, indicative of fully functional mitochondria. The combination of a fine morphological evaluation together

with the identification of specific mitochondrial markers enabled us to distinguish between clearing MDVs and small yet functional mitochondria.

However, other MDSs exhibited different staining patterns, suggesting the presence of multiple subgroups possibly serving different functions. Staining with the lipophilic dye, Nile Red, revealed labeling of both larger mitochondrial tubules and MDSs. A particularly

abundant (>70%) subset of vesicles and sticks colocalized with the peroxisome biogenesis factor PEX14. To elucidate the nature of these MDSs, we employed a sequential labeling strategy compatible with STED imaging to detect newly translated proteins, allowing for the classification of the age of MDSs directly in cells. These experiments unveiled that protein turnover occurs more rapidly at the cell soma, compared to distal filaments. Furthermore, within the PEX14 positive vesicles, the smaller ones (70–150 nm) lacked membrane potential but carried newly translated proteins. Mitochondria and peroxisomes share crucial metabolic functions, such as fatty acid oxidation and peroxide reduction, as well as some structural components, like DRP1 and the mitochondria fission factor (MFF)[39,40]. Recent studies have also implicated mitochondria in the de novo peroxisome biogenesis process[38]. In line with these studies, our findings suggest that MDSs contribute to lipid and protein inter-organelle trafficking, thus playing a pivotal role as MDVs in de novo peroxisome biogenesis in neuronal cells. This process involves not only MDVs but also ER-derived vesicles. However, the precise location of fusion between mitochondrial- and ER-derived vesicles to initiate pre-peroxisome maturation remains unclear[37]. By labeling different markers for pre- or mature peroxisomes, measuring their sizes, and assessing their subcellular location relative to mitochondria-ER contacts, we propose that fusion between ER-derived vesicles and MDVs occurs at mitochondrial sites. Moreover, the presence of both ER and DRP1 at the site of pre-peroxisome vesicle release suggests the involvement of the mitochondrial fission machinery in the process of de novo peroxisome biogenesis. Finally, downregulation of key regulators of pre-peroxisomes de novo formation, such as PEX3[38], or perturbation of mitochondrial fission machinery via expression of a dominant-negative DRP1K38A mutant, disrupted peroxisomes and specifically altered pre-peroxisomal structures associated with mitochondria. These findings indicate that DRP1-driven mitochondrial dynamics are required for establishing mitochondria–pre-peroxisome interactions and suggest that DRP1 and the ER coordinate the emergence of MDVs to initiate de novo peroxisome biogenesis.

In summary, STED nanoscopy enabled us to uncover the ultra-structure, composition, and age of MDVs in living neurons, facilitating the identification of their diverse roles in MQC and organelle biogenesis (Fig. 10). Given our evidence that mitochondria are the site of pre-peroxisome generation, where key players of mitochondrial fission are also found, further studies focusing on the 3D ultrastructural imaging at the ER-mitochondria contact-site can provide new insights into the mechanism of membrane fusion and pre-peroxisome formation.

## Methods

### Primary hippocampal neuron cultures

All experiments were performed in accordance with animal welfare guidelines set forth by Karolinska Institutet and were approved under the Ethical Permit nr 2645-2021 by the Swedish Board of Agriculture (Jordbruksverket).

Rats were housed with food and water available *ad libitum* in a 12-h light/dark environment. Primary hippocampal cultures were prepared from embryonic day 18 (E18) Sprague Dawley rat embryos. Embryos were collected randomly from the pregnant rats without prior determination of sex. The pregnant mothers were sacrificed with $CO_2$ inhalation and aorta cut; brains were extracted from the embryos. Hippocampi were dissected and mechanically dissociated in Minimum Essential Medium (MEM) (Thermo Fisher Scientific, 21090022). $40 \times 10^3$ cells per well were seeded in 12-well plates on a poly-D-ornithine (Sigma-Aldrich, P8638) coated #1.5 18 mm glass coverslips (Marienfeld, 0117580) and let them attach in MEM with 10% Horse Serum (Thermo Fisher Scientific, 26050088), 2 mM L-Glut (Thermo Fisher Scientific, 25030-024) and 1 mM Sodium pyruvate (Thermo Fisher Scientific, 11360-070), at 37 °C at an approximate humidity of 95–98% with 5% $CO_2$. After 3 h the media was changed to Neurobasal Medium (Thermo Fisher Scientific, 21103-049) supplemented with 2% B-27 (Thermo Fisher Scientific, 17504-044), 2 mM l-Glutamine, and 1% Penicillin-Streptomycin (Sigma-Aldrich, P4333). The cultures were kept at 37 °C at an approximate humidity of 95–98% with 5% $CO_2$ for up to 24 days. The medium was changed twice per week. The experiments were performed on cultures from DIV 5 up to DIV 10.

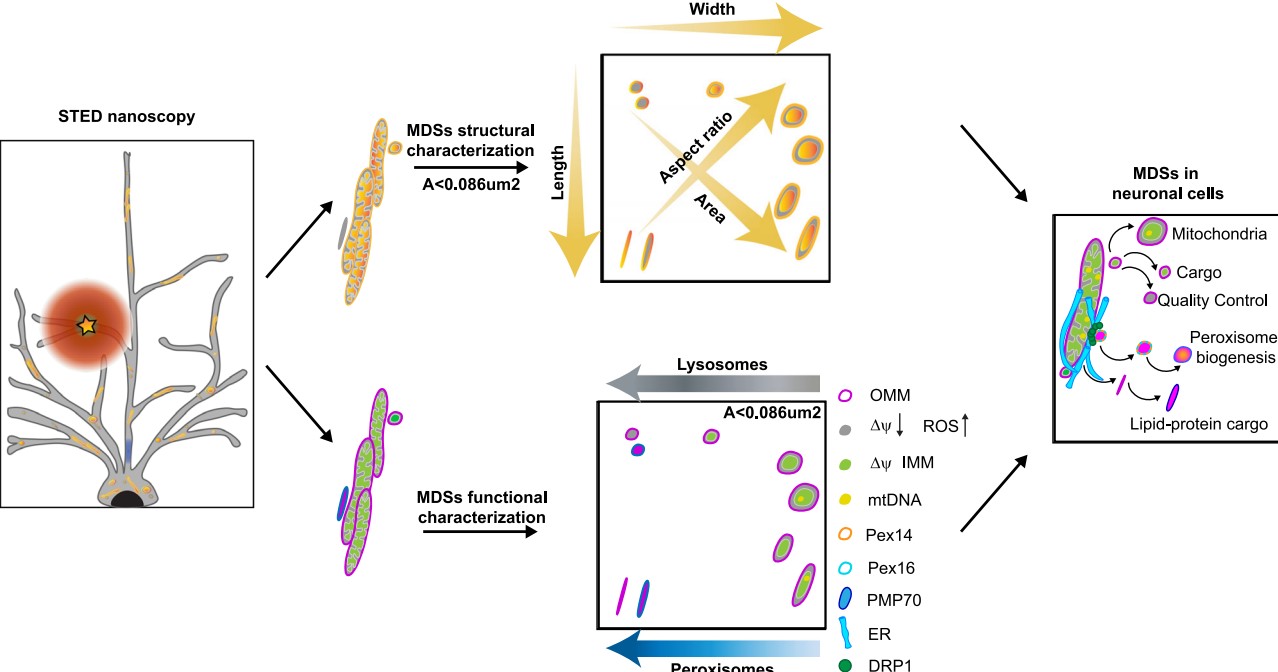

**Fig. 10 | Graphical illustration of the structural and functional diversity of mitochondrial-derived structures in neurons.** Multicolor STED nanoscopy reveals that in neuronal cells, MDSs comprise structurally and functionally distinct populations that mediate quality control, transport stress-related cargo, support local organelle and peroxisome biogenesis, and exhibit diverse morphologies and molecular signatures.

## Neuronal transfection and live-cell staining

Neurons were transfected between DIV 5 and DIV 10 using Lipofectamine 2000 Transfection Reagent (Thermo Fisher Scientific, 11668019), according to the instructions of the manufacturer.

Plasmids used: OMP25-rsEGFP2[27]; Halo-OMP25[28]; SNAP-Sec-61β[28]; SNAP-OMP25[23]; PEX16-rsEGFP2[50]; mCherry-DRP1[31] was a gift from Gia Voeltz (Addgene plasmid # 49152; http://n2t.net/addgene:49152; RRID:Addgene_49152); mCherry-DRP1K38A[44] was a gift from Samantha Lewis (Addgene plasmid # 232350; http://n2t.net/addgene:232350; RRID:Addgene_232350); Lamp1-RFP[51] was a gift from Walther Mothes (Addgene plasmid # 1817; http://n2t.net/addgene:1817; RRID:Addgene_1817); PEX14-GFP was a kind gift from the Eggeling Lab. To generate SNAP-tagged TFAM and COX8A, the plasmid pAAV-EF1a-mCherry-IRES-WGA-Cre[52], a gift from Karl Deisseroth (Addgene plasmid #55632; http://n2t.net/addgene:55632; RRID:Addgene_55632), was digested with BamHI and EcoRI. SNAP tag was inserted as a BamHI-EcoRI fragment, and it was amplified with the following oligos:

BamHI-SNAP-F atgactggatccGACAAAGACTGCGAAATGAAGCG
EcoRI-SNAP-R agtcatgaattcTTAACCCAGCCCAGGCTTGCC

The obtained pAAV_002-EF1a-SNAP plasmid was further digested with KpnI and BamHI. The CDS sequences of TFAM and COX8A from rat (NM_031326; NM134345, respectively) were inserted as KpnI-BamHI fragments. TFAM was amplified with the following oligos:

| KpnI-TFAM-F | atgactggtaccggattggccaccATGGCGCTGTTCCGGGGAAT |
| --- | --- |
| BamHI-TFAM-R | agtcatggatccgctaccgctgccATTCTCAGAGATGTCTCCCGG |

COX8A was amplified with the following oligos:
KpnI-COX8A-F
atgactggtaccggattggccaccATGTCTTCCCTGACGCCGC
BamHI-COX8A-R
agtcatggatccgctaccgctgccCTCCCGCTTCTTGTAGGTTTC

For labeling organelles fused with Halo-Tag and SNAP-Tag proteins, 24 h after transfection, neurons were washed in artificial cerebrospinal fluid (ACSF) and labeled with 200 nM of the SNAP or Halo substrates (New England BioLabs, SNAP-Cell 647-SiR, 590CP-Halo-ligand) for 1 h at 37 °C. Then, the neurons were washed three times with ACSF for 5 min each time. The cultures were washed again in ACSF and then either processed for immunostaining or imaged in the same buffer at room temperature (RT). For labeling the axon initial segment (AIS), neuronal cultures were incubated with anti-pan-neurofascin primary antibody (UC Davis/NIH NeuroMab Facility clone A12/18, 1:100), for 5 min at RT, subsequently washed three times with ACSF buffer, then incubated with anti-mouse-Alexa488 secondary antibody (Thermo Fisher Scientific, A-11001) for 5 min at 37 °C and finally washed three times with ACSF buffer.

For labeling mitochondria membrane potential, Tetramethylrhodamine Ethyl Ester Perchlorate (TMRE) (Thermo Fisher Scientific, T669) was used alone or in combination with MitoTracker™ Green FM (Thermo Fisher Scientific, M7514). The cells were loaded with TMRE at a final concentration of 100 nM, incubated at 37 °C for 1 h, and then imaged in ACSF media. When MitoTracker™ Green FM was required, cells were loaded with 20 nM of the dye for 15 min at 37 °C, shortly before imaging.

To label mitochondria ROS, the fluorescent indicator MitoSOX (Thermo Fisher Scientific, M36008) was added to the neurons at a concentration of 200 nM and left for 20 min in incubation. The cells were washed two times with ACSF and then imaged. To stain all cellular membranes, neurons were first washed in ACSF and then imaged in the same buffer supplemented with 300 nM of the lipophilic dye Nile Red (dissolved in MeOH) (Sigma-Aldrich, 72485). For cardiolipin-specific staining, 10-N-nonyl acridine orange (NAO) (Thermo Fisher Scientific, A1372) was added to the cell at 10 nM concentration for 30 min at 37 °C.

## siRNA

Three siRNAs (s222855; s222856; s222857) were purchased from Thermo Fisher Scientific and used to transfect neuronal cells using Lipofectamine™ RNAiMAX Transfection Reagent (Thermo Fisher Scientific, 13778100), following the manufacturer's instructions. The downregulation of PEX3 protein was monitored at 24 and 48 h after transfection via immunostaining with a specific PEX3 antibody (HPA042830, Atlas Antibodies).

## Immunofluorescence staining

Neurons were washed in ACSF and then fixed in pre-warmed 4% paraformaldehyde (PFA) in phosphate-buffered saline (PBS) (pH 7.4) at RT for 15 min and permeabilized for 5 min in 0.1% Triton-X-100 buffer, followed by a blocking step with 5% bovine serum albumin (A8022, Sigma-Aldrich) in PBS, for 30 min at RT. Incubation with primary and secondary antibodies was performed in PBS solution for 1 h at RT. Samples were mounted in custom-made Mowiol mounting media, supplemented with DABCO (Thomas Scientific, C966M75).

Antibodies and nanobodies are listed and used as follows: anti-PEX14 (Biosite, 10594-1-AP-20, 1:200); Total OXPHOS Rodent WB Antibody Cocktail (Abcam, ab110413, 1:200); anti-TOM20 (Santa Cruz Biotechnology, sc-11415, 1:50 dilution); anti-MAP2 (Abcam, ab5392, 1:2000); anti-TFAM (Abcam, ab13160, 1:200); FluoTag®-X4 anti-GFP (NanoTag, N0304-Ab580 or N0304-Ab635P-L); anti-PMP70 (Abcam, ab85550, 1:200); anti-mCherry (Abcam, ab167453, 1:200); anti-Drp1 (Cell Signaling Tech, D6C7 #8570); anti-PEX3 (Atlas Antibodies, HPA042830); anti-rabbit Alexa594 (Thermo Fisher Scientific, A-11037, 1:200 dilution); anti-mouse Alexa594 (Thermo Fisher Scientific, A-21203, 1:200 dilution); anti-rabbit STAR RED (Abberior, 2-0012-011-9, 1:200 dilution); anti-mouse STAR RED (Abberior, 2-0002-011-2, 1:200 dilution); anti-mouse AlexaFluor488 (Thermo Fisher Scientific, A-11001, 1:200 dilution), anti-Chicken AlexaFluor488 (Abcam, ab150173, 1:200 dilution).

## Drugs and metabolic treatments

To inhibit mitochondria respiration and specifically inhibit mitochondrial electron transport chain complex III and favors ROS production, antimycin A (Sigma-Aldrich, A8674) was used, following two different protocols: to induce a strong mitochondria stress (strong stress), the drug was added to the cells for 1 h at 40 μM or 50 nM final concentration; to induce a milder but prolonged stress (mild stress), neurons were cultured for 6 h in the presence of 5 nM of the same drug. After treatments, neurons were washed in ACSF and placed in fresh media at 37 °C for 1 h to equilibrate, before the imaging.

To determine the colocalization between MDSs and Lysosomes, in both control and treated samples, an automated decision was made through a MATLAB script using the binary mitochondria and lysosome images. The script measures the average Lysosome signal in each detected mitochondrion in the binary image. It then finds the threshold Lysosome value by plotting the histogram of the average Lysosome signal for all the mitochondria in an image, and fitting the histogram counts with an exponential decay (background) + Gaussian (true signal). The local minimum between the two peaks of the extracted function, i.e., the interception of the two curves, is taken as the threshold Lysosome value to which the individual mitochondria average Lysosome signal is compared. The threshold decision is done on an image-to-image basis, as different images have different signal intensities due to varying image acquisition parameters and labeling efficiency.

## MDS trafficking and turnover

MDS trafficking and turnover measurements were performed with a SNAP tag pulse-chase approach. In detail: neurons were transfected

with a SNAP-tag plasmid; for the control experiments, 0, 4, 13, 16, 18, 20 and 24 h after transfection ($T = 0$) cells were loaded with SNAP-Cell® TMR-Star ligand (NEB, S9105S) for 1 h at 3 µM; subsequently washed 3 times in ACSF; let equilibrate in fresh media for 1 h; fixed in 4% PFA for 20 min and immunostained for Map2, used here as a neuronal marker. For the turnover experiment, 24 h after transfection ($T = 0$), cells were loaded with SNAP-Cell® TMR-Star ligand for 1 h at 3 µM to label all proteins that were produced until that time; subsequently washed 3 times in ACSF; let equilibrate in fresh media for 1 h, and then let them in the incubator in fresh media for 24 h ($T = 1$). After this, cells were loaded with a second dye, SNAP-Cell® 647-SiR (NEB, S9102S) for 1 h at 200 nM, following the same washing procedure as before. The cells were then fixed in 4% PFA for 20 min and imaged in a confocal tile recording modality. This way, both old (SNAP-Cell® TMR-Star) and new (SNAP-Cell® 647-SiR) OMP25 were identified.

To image the new mitochondrial membrane and measure the mitochondria membrane potential at the same time, a similar approach was used, but instead of SNAP-Cell® TMR-Star, SNAP cell-block (NEB, S9106S) was added to cells for 1 h at 10 µM concentration, to block all the SNAP-tag proteins expressed at $T = 0$. 24 h later ($T = 1$) cells were incubated with 200 nM SNAP-Cell® 647-SiR and TMRE (Thermo Fisher Scientific, T669), at 100 nM concentration. After 1 h, the cells were washed, as above, and directly imaged in ACSF. This way, the newly produced proteins were first localized by SiR-647 fluorescence, and the membrane potential was further investigated by recording the TMRE signal. For the experiments on membrane refreshment in connection to peroxisomes, the same approach was used, and after fixation, the PEX14 antibody was further used to detect peroxisomes and investigate the presence of new OMP25 membranes at peroxisome sites, possibly indicating peroxisome de novo biogenesis.

## STED imaging and acquisition settings

STED nanoscopy has been performed at two different setups; a custom-built 3D-STED and a commercial Leica TCS SP8 3X STED. The custom-built STED setup is previously described in detail[53]. Briefly, excitation of the dyes was done with pulsed diode lasers: one at 561 nm (PDL561, Abberior Instruments), one at 640 nm (LDH-D-C-640, Pico-Quant), and one at 510 nm (LDF-D-C-510, PicoQuant). A laser at 775 nm (KATANA 08 HP, OneFive) was used as the depletion beam. The laser beams were focused on the sample using an HC PL APO 100×/1.40 Oil STED White objective (15506378, Leica Microsystems), through which also the fluorescence signal was collected. The imaging was done with a 561 nm excitation laser power of 8–20 µW, a 640 nm excitation laser power of 4–10 µW, and a 775 nm depletion laser power of 128 mW, measured at the first conjugate back focal plane of the objective. Both one and two-color STED imaging were done in a line-by-line scanning modality. The pixel size for all images was 20–30 nm. The pixel dwell time was 50 µs. The commercial Leica TCS SP8 3X STED is equipped with a HC PL APO 100×/1.40 Oil STED White objective. The images were recorded by exciting 590-CA/Alexa594 and 647-SiR/STAR RED with 590 nm and 650 nm laser lines, respectively. A STED beam at 775 nm has been used to deplete both laser lines. Two-color STED images were recorded line-by-line, averaging over 16 or 32 lines. The spectral detection windows were adjusted to 600–645 nm for 590-CA/Alexa594 and to 665–750 nm for 647-SiR/STAR RED.

## Mitography: automated mitochondria morphology data analysis

The automated image analysis of mitochondria and MDV morphology, Mitography, is carried out in two parts. Image processing and data gathering are performed in Fiji/ImageJ, while data handling, fitting, and visualization are performed in MATLAB (The Mathworks) (Supplementary Fig. 1i). This analysis can further analyze the position of individual mitochondria in relation to other fluorescent markers or neuronal compartments.

The ImageJ Mitography scripts take input images of mitochondria in the neurites of neurons labeled through the OMM (e.g., OMP25). Input images can be raw, smoothed, or deconvolved, where the smoothing or deconvolution can help with the robustness of the analysis and likely result in an increased number of successfully extracted data points. Tests have been performed to ensure that deconvolution or smoothing is not affecting the overall distribution of extracted and fitted morphological parameters. All morphology analysis presented here is performed on Richardson-Lucy deconvolved images. Prior to running the ImageJ Mitography analysis script, a binary mask of the mitochondria must be extracted. This is performed through thresholding and binarization steps, and depending on the labeling type, signal level, and uniformity of the structure in the field of view, different types of binarization are used. Provided in the scripts are two variants: one uses global thresholding, and the other mainly uses local Bernsen thresholding (local radius <10 pixels, contrast threshold <15). The thresholding is followed by morphological filtering (e.g., erosion, dilation, filling holes) to create a binary mask of the mitochondria and MDVs that is well-fitted to the actual size and shape of the mitochondria in the original image. Additionally, an inverted soma mask is extracted and used in the analysis when the soma is present in the image, to leave only mitochondria in the neurites for the analysis. In the main Mitography script, a well-dilated mitochondria binary mask is multiplied with the original OMM image, removing the background outside the mitochondria, and thus allowing for a better line profile extraction and subsequently a more robust fitting. The binary mitochondria image itself is also used initially for parameter extraction. *Analyze particles* and corresponding ellipsoidal fitting in Fiji is used to gather data such as ellipse centroid position, ellipsoidal length (ellipse major axis length), ellipsoidal width (ellipse minor axis length), area, and positioning angle (ellipse angle). To get a measure of the mitochondrial length that is more accurate for non-ellipsoidal-shaped mitochondria, a skeletonization is performed on the binary mitochondria representation. The *Analyze skeleton* plugin[54] and the summation of the branch lengths of the individual skeletons result in a second measurement of the mitochondrial length. For small mitochondria ($A < 0.2 \, \mu m^2$), the ellipsoidal length is always chosen, as it is more accurate for smaller mitochondria that are non-branched, non-curved, and where the skeleton does not cover the full length, while the longer of the two measurements is taken for larger mitochondria.

Line profiles (0.75 µm long, 5 pixels wide) are drawn across all binary mitochondria, in a direction perpendicular to the positioning angle, at three different points: the center and close to the endpoints along the long axis of the mitochondria. For shorter mitochondria ($L < 2.1 \, \mu m$), these points are chosen to be at $L/6$ and $5 \times L/6$, while for longer mitochondria ($L \geq 2.1 \, \mu m$), the points are chosen at 350 nm and $L - 350$ nm. The full set of extracted data and line profiles per mitochondrion are matched to the numbered mitochondria in the binary image.

In MATLAB, extracted parameters, line profiles, and binary maps are imported. Here, line profiles of mitochondria are fitted to extract the width at the center and outer positions. Firstly, the number of peaks of the line profile is extracted. If the number of peaks is 1 or 2, a fitting is performed with a combination of Gaussian functions, either as a single (no outer membrane visible, as in confocal images of most mitochondria and STED images of narrow MDVs) or double (outer membrane visible, as in STED images of most mitochondria and confocal images of wide mitochondria). While fitting a single-Gaussian function across two unresolved membranes will overestimate the width of the mitochondrion, it is the best estimation of the width that can be extracted from such a case. Furthermore, disregarding these cases entirely would result in a far larger shift of the overall distribution of widths. If three or four peaks are located, it is likely that we have two mitochondria side-by-side with outer membranes visible. In this case, the positions of the peaks are used to calculate the distance between

the individual peaks, and these values are used as the mitochondria widths. These mitochondria are generally disregarded later in the analysis, due to a more uncertain and likely erroneous parameter extraction for non-width parameters, but they can be investigated further if of interest. If more than four peaks are found in the line profile, or the R-square values of the fits are below set thresholds, the mitochondrion is immediately disregarded due to any parameter extraction being far too erroneous, and the mitochondrion likely not being accurately detected. In most cases, however, the FWHM of the single-Gaussian fit in the case of a single detected peak, or the distance between the two peaks of the double-Gaussian fit in the case of two detected peaks, is extracted as the mitochondrial width, assuming the R-square value of the respective fit is higher than a set threshold. Following the fitting, the center position of each mitochondrion is checked against the provided binary images of interest (e.g., dendrite marker, axon initial segment marker). The binary pixel value of each position is saved as a Boolean variable for each mitochondrion, generating a register of Boolean parameters defining the position of the mitochondria with respect to various neuronal compartments. Additionally, secondary morphological parameters are also calculated from the extracted morphological parameters. These include the AR, $L/W_m$, and the width ratio between the outer and center width, $W_o/W_m$. All extracted parameters and fitting parameters, per mitochondrion, are saved in a tab-delimited text file.

## Automatic TMRE and MitoSOX $+/-$ detection

To determine if each mitochondrion is $TMRE^+/MitoSOX^+$ or $TMRE^-/MitoSOX^-$, an automated decision was made through a MATLAB script using the previously extracted binary mitochondria image and the TMRE/MitoSOX image. The script measures the average TMRE/MitoSOX signal in each detected mitochondrion in the binary image. It then finds the threshold TMRE/MitoSOX value by plotting the histogram of the average TMRE/MitoSOX signal for all the mitochondria in an image, and fitting the histogram counts with an exponential decay (background) + Gaussian (true signal). The local minimum between the two peaks of the extracted function, i.e., the interception of the two curves, is taken as the threshold TMRE/MitoSOX value to which the individual mitochondria average TMRE/MitoSOX signal is compared. The threshold decision is done on an image-to-image basis, as different images have different signal intensities due to varying image acquisition parameters and labeling efficiency.

## Mitochondria processes number density analysis

To calculate the mitochondria number density in processes, the lengths of the processes were needed. These were found by using the background fluorescence in the images of the outer mitochondrial membrane labeled with the OMP25 localization peptide, and this was done in a Python script. The image was processed by removing the mitochondria in the soma through multiplication with a binary soma map, setting all pixels above a threshold to the threshold value, performing a global binarization of the image based on a threshold calculated with the triangle method, performing morphological operations such as removing small objects (<20 pixels), dilation (20×), and erosion (15×), and removing everything except the largest connected segment, a binary map of the processes was found. This was skeletonized, and by summing the length of all the skeletal branches, the total length of the processes for the image was found. Each image contained either an axon or dendrites (a decision based on an image of, for example, map2 labeling), and hence the total number density for axons and dendrites could be found on an image-to-image basis. The *triangle method*, morphological operations, and *skeletonization* were all performed using the implementations in the Python package scikit-Image[55].

## Quantification of MDS trafficking and turnover dynamics

Quantification of the turnover dynamics was performed by a combination of mitochondrial morphology through the above-described analysis scripts and analysis of the signal of the two labels from different timings for each mitochondrion. The refreshment rate was calculated as the sum of the signal from SNAP-Cell® SiR-Star ligand inside the mitochondrial area, divided by the sum of the signal from SNAP-Cell® TMR-Star ligand and the sum of the signal from SNAP-Cell® SiR-Star ligand inside the mitochondrial area. This ratiometric rate takes values between 0 and 1, where 1 indicates 100% refreshed protein and 0 indicates 0% refreshed protein in the investigated time interval. In these experiments, the ratio of fluorescence signal does not increase linearly with the ratio of refreshed proteins, due to different properties of the dyes and different excitation powers, leading to different excitation and detection efficiencies. Hence, the ratiometric value cannot be translated directly into the ratio of refreshed proteins. An automatic radial analysis was performed, starting from the cell soma and moving toward the distal filaments, dividing the cell into concentric rings of 30 μm thickness. Quantification of the control turnover experiments was done by a combination of automated and manual analysis. Ilastik[56] was used to automatically segment the neuronal cell based on the MAP2 fluorescent signal, followed by a manual correction to remove small particles and closed holes, and to ensure that only one successfully transfected cell per image was displayed. The cell soma was further removed from the segmentation to focus the analysis on neuronal filaments exclusively. A mitochondrial binary map was made with Mitography. Subsequently, an automatic radial analysis was performed, starting from the cell soma and moving toward the distal filaments, dividing the cell into concentric rings of 30 μm thickness. The fluorescence intensity of mitochondrial structures throughout the cell was measured as the levels of SNAP-Cell® TMR-Star intensity, normalized by its intensity in the most proximal mitochondria to compensate for varying expression levels in different cells, for each ring, for four different time points: 0 h, (data not shown), 4–5 h, 13–18 h and 20–24 h.

## Manual mitochondrial protrusions quantification

Mitochondrial protrusions were selected manually, based on the OMP25 channel. To determine if the protrusion was "tip" or "side," the length of the mitochondrial tubule was divided into quarters: protrusions in the two quarters at the extremity of the tubules were considered "tip", while protrusions found in the two middle quarters were considered "side". To determine whether each protrusion was positive or negative (+/−) for the additional markers (PEX14, PEX16, PMP70, DRP1), fluorescence intensity was quantified at the corresponding position in each channel. The measured signal was then compared with background intensity, defined as the average fluorescence intensity within the neuronal filaments but outside any labeled structure. To access the ER association, fluorescence intensity was measured both at the same position overlapping the protrusion and along the protrusion boundary facing the mitochondrial side.

## Quantification of PEX3 RNAi silencing

PEX3 puncta were identified based on PEX3 immunofluorescence, and a binary mask was generated. The number and fluorescence intensity of PEX3 puncta, as well as the linear number density of mature peroxisomes (PMP70 immunostaining), were measured in cells transfected with siRNA for 48 h and compared with those in non-transfected cells.

## DRP1 analysis

DRP1WT and DRP1K38A transfected neurons were identified by the fluorescent signal of the mCherry fusion tag, and a binary mask was

generated to ensure that only transfected neurons were included in the analysis. PEX14 puncta and mitochondria were identified based on PEX14 and ATPb immunofluorescence, respectively, and corresponding binary masks were created. Mitochondrial length was quantified with Mitography, and to assess the ability of DRP1K38A to impair mitochondrial fission, the percentage of mitochondria longer than 10 µm was compared between DRP1WT and DRP1K38A expressing neurons. To investigate the effects of mitochondrial fission perturbation on peroxisome biogenesis, mitochondria-PEX14 or mitochondria-PMP70 contact sites were quantified in DRP1WT and DRP1K38A expressing cells.

## Image analysis, code availability, and statistical tests

Raw images were processed and visualized using the ImSpector software (Max-Planck Innovation, Gottingen, Germany) and Fiji/ImageJ[57,58]. When necessary, images and movies were deconvolved using the Richardson-Lucy algorithm, implemented in Imspector. The PSF was modeled as a Gaussian function with a FWHM of 40–60 nm. The regularization parameter ($10^{-5}$ or $10^{-10}$) and number of iterations (max 10) varied depending on the quality of the output image. Movies 1–3 are instead smoothed using a low-pass Gaussian filter, implemented in Imspector. When necessary, movies were corrected for bleaching using the "Bleach correction" plugin and for drift using the "MultiStackReg" (based on the "StackReg") plugin in Fiji. Plotted line profiles in the figures were averaged over 2–4 pixels and fitted with the software OriginPro2020. All automated analysis has been performed mainly in Fiji/ImageJ and MATLAB (The Mathworks), with some auxiliary data handling performed in Python. Code and scripts for the various parts of the analysis in ImageJ and MATLAB, also described above, together with example data for running the scripts, are available online at https://github.com/jonatanalvelid/mitography-public and as a versioned release in Zenodo at https://doi.org/10.5281/zenodo.17831561. See documentation in the readme and info files for further information on the specific scripts and how to run them. For comparison of distributions of parameters for different conditions, a Student's $t$-test (normally distributed data) or a Kolmogorov–Smirnov test (non-normally distributed data) was chosen.

## Reporting summary

Further information on research design is available in the Nature Portfolio Reporting Summary linked to this article.

## Data availability

The data supporting the findings of this study are available from the corresponding author upon request. Microscopy data reported in this paper will be shared by the lead contact upon request. Source data for the main figures are provided with this paper. Source data are provided with this paper.

## Code availability

All original data analysis code is available in the provided GitHub repository, as well as in the Zenodo-deposited version-specific release of the code at https://doi.org/10.5281/zenodo.17831561. Any additional information required to reanalyze the data reported in this paper is available from the lead contact upon request.

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

## Acknowledgements

The authors thank Francesca Pennacchietti and Michael Ratz for critical reading and guidance; the ALM facility for access to the STED SP8 Leica microscope; Katharina Reglinski and Delgir Zakinova (Eggeling Lab) for sharing the PEX14-GFP plasmid. I.T. thanks the ERC consolidator grant (InSpIRe-101002490) for supporting this research project.

## Author contributions

I.T. supervised the project. I.T. and G.C. designed the experiments. J.R. provided biological guidance. G.C. cultured and labeled neurons, optimized the labeling protocols, performed the experiments, and analyzed the data. J.A. built the STED optical setup, developed the software Mitography, and analyzed the data. M.D. carried out the cloning and neuronal preparation. G.F. performed the biogenesis experiments. J.M. helped with the data analysis of the biogenesis experiments. I.T. and G.C. wrote the manuscript with assistance from all the authors.

## Funding

## Competing interests

The authors declare no competing interests.
