## [Transparent Peer Review file · Nature Communications]

Quantitative optical nanoscopy of mitochondrial-derived vesicles in neurons classify pre-peroxisomal and clearing organelles.”.

Corresponding Author: Professor Ilaria Testa

Version 0:

Reviewer comments:

Reviewer #1

(Remarks to the Author)

The manuscript produced by Coceano et al. provides an excellent characterization of MDVs in neurons utilizing powerful microscopy techniques. However, data do not strongly support much of the functional roles described regarding these vesicles, and some major concerns brought up by the reviewers were not addressed. Unfortunately, because the current driving take-home narrative of the manuscript relies on these immature data, I am not confident that the state of this manuscript is ready for publication in Nature Communications. Further details can be found below:

Major Concerns:

1. Two reviewers had major concerns with the experimental setup and interpretation that MDVs in different neuronal compartments refresh their membrane content at different rates. However, the authors did not experimentally show that newly synthesized proteins in the Halo/Snap-tag system are equally integrated in somal vs distal organelles. The authors should experimentally address this concern or, at minimum, discuss alternative interpretations within the text.
 - a. There is also a statement about existing MDVs potentially incorporating newly synthesized OMP25-Snap, to which they provide a logical argument in lieu of experimental data that is potentially not grounded in current literature.
 - b. Since interpreting their results relies on understanding these dynamics, the authors should put pronounced effort into generating argument-supporting data.
2. In agreement with Reviewer 2, I am unclear of the novelty of peroxisomes and MDV biogenesis reported in the manuscript. Multiple studies describe MDV involvement in the de novo biogenesis of peroxisomes, providing mechanistic characterization (DOI: 10.1038/nature21375, and more recently DOI: 10.1016/j.devcel.2024.09.029). However, as stated by the authors, none of these studies were performed in neurons. Therefore, it is critical to do some characterization to support the novelty and arguments made within this study. For example, could Pex3 or another marker be depleted in neurons to look at overall effects? As the manuscript currently stands, this point represents a significant gap in the study.
3. The authors make claims about the functional roles of proteins and organelles in forming pre-peroxisomes. They make these statements almost entirely from localization data without perturbing key proposed players, as multiple reviewers requested (e.g., MDV biogenesis and Drp1). Additionally, they include exciting ER data visualizing tubules at MDV/pre-peroxisome scission sites in the updated manuscript. However, they do not acknowledge that these data are inconsistent with the model that ER vesicles fuse with the maturing pre-peroxisome.
 - a. They also generally do not follow through on investigating the fate of these mitochondrial protrusions through timelapse.
 - b. The representative images for ER and Drp1 are nice, but quantification is necessary to support the conclusions within the text.
 - c. Experiments highlighting the role of Drp1 through perturbations, such as Drp1 KD or overexpression of Drp1 mutants, were again suggested by reviewers but not conducted.

Minor Concerns:

1. The authors should include novelty and motivation for using neuronal cultures in the background, as suggested by Reviewer 2. This would differentiate the study and support their claims of neuronal novelty.

2. Reviewer 3's comments state that size cannot be used alone to determine an MDV, and the authors agree with the comments. However, the text was not altered to address these changes. Lines 121-123 still suggest using size to define an MDV.

3. The authors state that "only those attached to mitochondria and labelled with OMP25 were considered" for quantification in Figure 8. Could the authors state what percentage of vesicles were used in this quantification? If it is a small population, this is important to know for the interpretation of the data. It is unclear whether this minor concern was addressed based on the revised manuscript.

4. There are no error bars on the normalized frequency plots (example, Fig. 4G).

Reviewer #2

(Remarks to the Author)

The authors have done a fair job in addressing my previous comments, except for specific comment No. 4, which they could address with finding/acquiring pairs of images (confocal vs. STED) where the potential confounder of structural movement is absent.

Otherwise, I find the paper suitable for Nat. Communications.

Reviewer #4

(Remarks to the Author)

The concerns have been addressed, and the manuscript is well organized for publication.

Reviewer #5

(Remarks to the Author)

Reviewer #6

(Remarks to the Author)

Version 1:

Reviewer comments:

Reviewer #1

(Remarks to the Author)

The revised manuscript by Coceano et al. provides novel insights into mitochondrial-derived structures within neurons. The use of STED super-resolution microscopy provides a powerful platform to investigate structural characteristics. The authors have taken the reviewers' comments into account and provided new experiments and analyses that support their arguments and strengthen their functional model. Overall, this work will benefit the field, and I believe this manuscript is now ready for publication in Nature Communications.

Minor Comments:

- In Figure 9e-f, the authors mention "nearby" ER signal as part of their quantification criteria, but I do not see where this characteristic was defined. Was there a specific distance that the ER signal needed to fall within?
- Occasionally, there are grammatical errors in the new text (for example, Line 387 says "this mutant exhibit reduced efficiency" instead of "this mutant exhibits reduced efficiency"; Line 217, the word few is misspelled), so I recommend a quick grammar check before finalizing the paper.

Point by point response to Reviewers of the manuscript NCOMMS-24-82932-T:

Quantitative optical nanoscopy of mitochondrial-derived vesicles in neuronal cells classify pre-peroxisomal and clearing organelles

Coceano et al.

We thank the editor for the interest in our work and the reviewers for the critical assessment of our study which led us to further develop the connection between Mitochondrial-Derived Vesicles (MDVs) and peroxisome biogenesis in the revised version.

We performed new experiments 1- to support the pulse-chase experiments and 2- to provide new insights about the connection between MDVs and peroxisome biogenesis. The new data about mitochondrial membrane refreshment is summarized in the **new Supplementary Fig. 2**. Additionally, new experiments aimed at altering the process of peroxisome biogenesis are summarized in the **new Supplementary Fig. 8 and 9**.

We modified the text to explain and discuss the new findings, including the motivation for and the novelties of using neuronal cultures. We clarified in the text the different parameters used to identify and characterize MDVs in neuronal cells. The term “mitochondrial-derived structures (MDSs)” was chosen instead of “mitochondrial-derived vesicles (MDVs)” to characterize the diverse nano-sized structures, vesicular and non-vesicular, derived by mitochondrial outer membranes and here resolved by super-resolution light microscopy.

We updated Fig. 1, 4, 6, 8, 9 following the reviewer’s suggestions and modified the text accordingly.

We modified the text and figures to adhere to the Nature Communication formatting requirements.

Please find below a point-to-point response (blue) to the reviewers’ questions (black).

Reviewer #1 (Remarks to the Author):

The manuscript produced by Coceano et al. provides an excellent characterization of MDVs in neurons utilizing powerful microscopy techniques. However, data do not strongly support much of the functional roles described regarding these vesicles, and some major concerns brought up by the reviewers were not addressed. Unfortunately, because the current driving take-home narrative of the manuscript relies on these immature data, I am not confident that the state of this manuscript is ready for publication in Nature Communications. Further details can be found below:

Major Concerns:

1. Two reviewers had major concerns with the experimental setup and interpretation that MDVs in different neuronal compartments refresh their membrane content at different rates. However, the authors did not experimentally show that newly synthesized proteins in the Halo/Snap-tag system are equally integrated in somal vs distal organelles. The authors should experimentally address this concern or, at minimum, discuss alternative interpretations within the text.
 - a. There is also a statement about existing MDVs potentially incorporating newly synthesized OMP25-Snap, to which they provide a logical argument in lieu of experimental data that is potentially not grounded in current literature.
 - b. Since interpreting their results relies on understanding these dynamics, the authors should put pronounced effort into generating argument-supporting data.

We thank the reviewer for raising relevant questions and considering our work an excellent imaging characterization of MDVs in neurons.

As a follow up of the discussion initiated in the previous response letter, it is plausible to think that the plasmid expression and protein trafficking might influence the rate of newly versus elderly expressed proteins on the outer membranes of mitochondria and MDVs. However, we saw that vesicles trafficking in neuronal processes happens very fast (seconds-minutes) at this neuronal age and therefore these dynamics tend to be faster than the 24 hours interval between the pulse and chase experiments.

To shine more light into the reviewer's concerns and experimentally demonstrate that newly synthesized proteins are integrated both in proximal and distal organelles already during the first few hours of expression, we analyzed the rate of transient plasmid expression and correlated it to the spatial distribution of the OMM-SNAP fusion protein within the neuronal cell (Revision Fig. 1 and Supplementary Fig. 2 in the manuscript). These experiments aim to show when and where, along the processes of the neuronal cell, exogenous proteins are displayed. We expressed the exogenous OMM-SNAP plasmid into DIV8 neuronal cells and labelled the SNAP-tag fusion protein with TMR-Star-BG ligand at different time-points after transfection: 0, 4, 13, 16, 18, 20 and 24 hours (Revision Fig. 1a). We then analyzed the distribution of OMM-SNAP labelled structures and quantified their fluorescence intensity as a function of their distance to the cell soma. The results show that shortly after transfection (T=0 h) no TMR-Star signal can be detected within the neuronal cells. However, already 4 h after transfection, TMR-Star-BG signal was detected on mitochondria-like structures (Revision Fig. 1b). The fluorescence intensity was distributed throughout the neuronal cell, both in the soma and along neurites. Comparison of TMR-Star-BG intensity levels at different distance from the soma, for each cell normalized to the intensity of the most proximal mitochondria to account for differences in expression across cells, between different time intervals (4–5, 13–18, and 20–24 hours) revealed no significant variation of the distribution. The shaded areas represent 83.4% confidence intervals and thus their overlap corresponds to $p > 0.05$ based on Student's t-tests comparing the means (Revision Fig. 1c–d). These results demonstrate that the exogenous OMM-SNAP is integrated both in the soma and in the distal filaments already within the first 4 h of protein expression in the same way as after 24 h. While there is a gradient in TMR-Star intensity along the neurites, indicating that a higher expression level of OMM-SNAP might be present at proximal mitochondria, its consistency proves that the protein turnover measured after 24 h is not influenced by differences in expression between proximal and distal organelles at different timepoints. Taking together, these results indicate that OMM-SNAP turnover is higher in proximal mitochondria than in distal ones. This conclusion goes well along with the results shown in the original experiment (Main Text Fig. 2), where the measured intensity ratio between newly labeled and total (new + old) proteins, labeled with two spectrally distinct dyes, accounts for factors such as the observed expression gradient. We cannot, however, exclude the fact that the overall turnover rate measured here is not affected by the protein tag. To assign a correct value for a specific protein's turnover rate, which is currently outside the scope of this work, we should test the influence of the tag by comparing it with endogenous protein expression. We then should apply the experimental pulse and chase procedure developed here in a systematic way to measure the turnover rate of different mitochondrial proteins. We are currently working to test and compare the turnover rates of mitochondrial proteins from different mitochondrial sub-compartments: such as Tom20, for the outer mitochondrial membrane, TFAM, for mitochondrial nucleoids, and ATP synthase subunit beta, for the inner mitochondrial membrane. Preliminary data show different spatial distributions, but they are not mature enough to be added here since they lack a direct comparison with untagged proteins and can be part of a follow-up work. In the current work we are not quantifying the OMP25 turnover rate but rather using the OMM tagging strategy to compare membrane refreshment rates in different subcellular locations.

Regarding **the second point** mentioned by the reviewer and hence the possibility of newly translated proteins directly imported into existing MDVs, the reviewer probably refers to the work done by McBride and colleagues (DOI: 10.1038/s41556-021-00798-4), where they showed that MDVs can incorporate the full Tom protein import complex. In the mentioned paper, the authors proposed a new mechanism of quality control, based on Tom20 MDVs, which could help in removing fully assembled protein complexes for the final degradation in the lysosomal compartment. From literature data, we also know that vesicles targeting peroxisomes are negative for Tom20 and therefore these vesicles are different with respect to the ones involved in mitochondrial quality control. Following these findings, we therefore suppose that peroxisome-directed MDVs do not feature functional import machinery. Moreover, our findings show newly synthesized OMM-SNAP signal within vesicles colocalizing with peroxisome biogenesis markers, such as PEX14, and not Tom20, in potential agreement with the data available in literature about the involvement of MDVs in the *de novo* biogenesis of peroxisomes (DOI: 10.1038/nature21375).

Revision Fig. 1

(a) Workflow scheme of the SNAP-tag protein expression system to follow the exogenous protein distribution over time and space within the full neuronal cell. The SNAP-tag labelling of the protein of interest (POI) with SNAP-TMR-Star-BG ligand is performed at different timepoints after transfection (0, 4, 13, 16, 18, 20 and 24 h), followed by fixation and immunostaining of the protein Map2 (blue).

(b) Representative example of a neuronal cell transfected with OMM-SNAP plasmid for 4 hours, labelled with SNAP-TMR-Star-BG (OMM, green) and Map2 (blue).

(c) Workflow scheme of the image analysis for the quantification of the exogenous protein distribution over time and space

(d) Quantification of the SNAP-TMR-Star-BG intensity (OMM), normalized by its intensity in the most proximal mitochondria, measured at different distances from the cell soma, for three different transfection times: 4-5 h (grey); 13-18 h (yellow) and 20-24 h (green). Shaded areas denote 83.4% confidence intervals and thus their overlap corresponds to $p > 0.05$ for Student's t-tests for different means. Data were collected from 34 cells from 7 independent samples.

2. In agreement with Reviewer 2, I am unclear of the novelty of peroxisomes and MDV biogenesis reported in the manuscript. Multiple studies describe MDV involvement in the *de novo* biogenesis of peroxisomes, providing mechanistic characterization (DOI: 10.1038/nature21375, and more recently DOI: 10.1016/j.devcel.2024.09.029). However, as stated by the authors, none of these studies were performed in neurons. Therefore, it is critical

to do some characterization to support the novelty and arguments made within this study. For example, could Pex3 or another marker be depleted in neurons to look at overall effects? As the manuscript currently stands, this point represents a significant gap in the study.

As a follow up of the reviewer suggestion, we discussed the novelties of our study in more detail within the manuscript, specifically concerning the importance of MDV characterization and related peroxisome biogenesis in neuronal cells. We also performed the additionally suggested experiment to deplete PEX3 in neuronal cells and quantified the overall effects (Revision Fig. 2 and Supplementary Fig. 8 in the manuscript).

To deplete the expression of PEX3 in neuronal cells, different strategies can be used. Among them, we choose RNA silencing as a strategy to transiently downregulate the amount of PEX3 protein to alter the process of peroxisome *de novo* biogenesis. We selected three commercially available siRNA, targeting three different regions of the PEX3 mRNA transcript (s222855; s222856; s222857 ThermoFisher Scientific). Following the manufacturer instructions, we transfected the siRNAs into DIV6 neuronal cells and checked the downregulation of PEX3 protein 24 and 48 h after transfection via immunostaining with a specific PEX3 antibody (HPA042830, Atlas Antibodies). The results, summarized in Revision Fig. 2 and Supplementary Fig. S8 in the main text, show that upon RNA silencing there is an overall reduction in the amount of PEX3 in treated cells with respect to control untreated cells. Specifically, we measured the intensity of PEX3 punctae in neuronal cells transfected with siRNA for 48 h and compared it with those in non-transfected cells. The fluorescence intensity of labelled endogenous PEX3 is significantly lower in transfected with respect to non-transfected neuronal cells (Revision Fig. 2 a-c, two-sample t-test $p = 7.68462 \cdot 10^{-4}$). These results confirmed the efficient reduction of PEX3 mRNA translation upon transient RNA silencing. Then, we quantified the number of PEX3 punctae in treated and control samples and we observed a reduction in their linear number density when PEX3 mRNA was silenced (two-sample t-test $p = 0.01075$) (Revision Fig. 2d). Moreover, the same results were obtained when quantifying the linear number density of mature peroxisomes, labelled here with the peroxisomal membrane protein 70 (PMP70) (two-sample t-test $p = 0.00903$) (Revision Fig. 2 e-g). Together, these findings demonstrate that transient PEX3 knockdown effectively impairs the *de novo* peroxisome formation in neuronal cells, as reflected by the reduced abundance of both PEX3-positive and mature peroxisomal structures.

Revision Fig. 2

(a) Representative example of a neuronal cell where exogenous PEX3 proteins are fluorescently labelled via immunostaining with an anti-PEX3 specific antibody. ROI I shows a neuronal filament where PEX3 punctae are visualized in the confocal scan (upper panel) and subsequently masked based on their intensity (bottom panel).

(b) Representative example of a neuronal cell, treated for 48 h with 3X siRNAs molecules to downregulate the translation of PEX3, and fluorescently labelled via immunostaining with an anti-PEX3 specific antibody. ROI II shows a neuronal filament where PEX3 punctae are visualized in the confocal scan (upper panel) and subsequently masked based on their intensity (bottom panel).

(c) Quantification of the PEX3 fluorescence intensity measured in control (grey) and in siRNA treated cells (cyan). Two-sample t-test $p = 7.68462 \cdot 10^{-4}$.

(d) Quantification of the PEX3 number density measured in control (grey) and in siRNA treated cells (cyan). Two-sample t-test $p = 0.01075$.

(e) Representative example of a neuronal cell where exogenous PMP70 proteins are fluorescently labelled via immunostaining with an anti-PMP70 specific antibody. ROI III shows a neuronal filament where PMP70 punctae are visualized in the confocal scan (upper panel) and subsequently masked based on their intensity (bottom panel).

(f) Representative example of a neuronal cell, treated for 48 h with 3X siRNAs molecules to downregulate the translation of PEX3, and fluorescently labelled via immunostaining with an anti-

PMP70 specific antibody. ROI IV shows a neuronal filament where PMP70 punctae are visualized in the confocal scan (upper panel) and subsequently masked based on their intensity (bottom panel).

(g) Quantification of the PMP70 number density measured in control (grey) and in siRNA treated cells (cyan). Two-sample t-test $p = 0.00903$.

3. The authors make claims about the functional roles of proteins and organelles in forming pre-peroxisomes. They make these statements almost entirely from localization data without perturbing key proposed players, as multiple reviewers requested (e.g., MDV biogenesis and DRP1). Additionally, they include exciting ER data visualizing tubules at MDV/pre-peroxisome scission sites in the updated manuscript. However, they do not acknowledge that these data are inconsistent with the model that ER vesicles fuse with the maturing pre-peroxisome.

a. They also generally do not follow through on investigating the fate of these mitochondrial protrusions through timelapse.

b. The representative images for ER and Drp1 are nice, but quantification is necessary to support the conclusions within the text.

c. Experiments highlighting the role of Drp1 through perturbations, such as Drp1 KD or overexpression of Drp1 mutants, were again suggested by reviewers but not conducted.

We performed new experiments to support the functional roles of mitochondrial-derived structures in the process of pre-peroxisome formation.

a. We performed time-lapse imaging of mitochondria and peroxisomes to identify mitochondrial protrusions that colocalize with peroxisomal biogenesis markers, track their detachment and follow the formation of separate MDSs. Mitochondria were labelled with OMM-Halo to visualize membrane protrusions, while PEX14-GFP was expressed to label peroxisomes. Confocal and STED time-lapse imaging were used to monitor the formation of mitochondrial protrusions colocalizing with PEX14-GFP and to follow their detachment from the mitochondrial tubule, resulting in the formation of mitochondrial-derived structures that colocalize with the peroxisomal marker (Revision Fig. 3a-b; Supplementary Fig. 9 a-b).

b. We modified Fig. 9 according to the reviewer concerns and specifically added the quantification for the identified protrusions. The results overall show that most of the analysed mitochondrial protrusions have ER signal nearby when they are colocalizing with either PEX16 or PEX14. With respect to DRP1, we show that ER and DRP1 are both present in most of the identified mitochondrial protrusions, suggesting their involvement in the process of mitochondrial membrane protrusion formation; moreover, DRP1 was also detected at most mitochondrial protrusions colocalizing with PEX14, pointing to its involvement in the initial steps of the process of the *de novo* peroxisome biogenesis.

Following the process of vesicle formation by timelapse imaging and visualizing ER structures at the site of vesicles release resemble the process happening during mitochondrial fission, where ER tubules wrap around the site of constriction to favour the tubule fission, and therefore suggest that ER could potentially provide the same support also in the case of MDS formation. Moreover, the proximity of ER at mitochondrial protrusions could facilitate the encounter between mitochondria and ER-derived vesicles, a condition required for the *de novo* peroxisome biogenesis process.

c. We performed new experiments to perturb DRP1 function through overexpression of the DRP1K38A mutant, which harbours a dominant-negative mutation in the guanosine triphosphate (GTP)-binding pocket, thereby inhibiting GTP binding and hydrolysis. Previous studies have demonstrated that this mutant exhibits reduced efficiency in mediating mitochondrial constriction and division (DOI:10.1091/mbc.12.8.2245) in the new

Supplementary Fig. 9c-i and Revision Fig. 3, show that DRP1K38A overexpression induces the formation of elongated mitochondrial tubules and promotes an overall elongation of the mitochondrial network relative to cells expressing wild-type DRP1 (DOI: 10.1126/sciadv.ads6830) (Revision Fig. 3c-e). Analysis of the spatial relationship between mitochondria and pre-peroxisomal markers, such as PEX3 and PEX14, revealed a significant reduction in the number of pre-peroxisomal punctae contacting mitochondria in neuronal cells expressing DRP1K38A compared to control cells, as measured by the linear number density of contact sites per mitochondrial length (Revision Fig. 3 f-h). In contrast, no differences were observed in the contact sites between mitochondria and mature peroxisomes labelled with PMP70 (Revision Fig. 3i). Collectively, these findings indicate that disruption of DRP1 function compromises mitochondrial dynamics and selectively impairs the association between mitochondria and pre-peroxisomal structures, without affecting mature peroxisome interactions. These observations further suggest a potential role for DRP1 in the regulation of de novo pre-peroxisome biogenesis mediated by mitochondrial-derived structures.

Revision Fig. 3

(a) Representative two-color live confocal time-lapse imaging of mitochondria (Halo-OMM, magenta) and peroxisomes (PEX14, yellow). Zoom-in of five frames, shown as single-color for

mitochondria (upper line) and peroxisomes (bottom line), showing the formation and detachment of a mitochondrial protrusion colocalizing with PEX14 at the tip of a mitochondrial tubule. The last frame is shown as a two-color image with the line profile, measured along the denoted line in the panel, at the detachment site. The movie has been recorded for roughly 2 minutes at a frame rate of 6 frames/minute.

(b) Two more examples of mitochondrial protrusions colocalizing with PEX14 and the formation of an MDV followed in confocal (left side, roughly 1.5 minutes at a frame rate of 6 frames/minute) and STED (right, roughly 30 seconds at a frame rate of 6 frames/minute) timelapse imaging.

(c) Two representative examples of the mitochondrial network (ATPb immunostaining) in a neuronal cell exogenously expressing DRP1WT (left panel, mCherry-DRP1WT is shown in the inset) or DRP1K38A mutant (right panel, mCherry-DRP1K38A is shown in the inset). Different from DRP1WT, DRP1K38A overexpression induces the formation of elongated mitochondria.

(d) Distribution of the mitochondrial lengths measured in neurons expressing DRP1WT (grey) or DRP1K38A (cyan). The data were collected from DIV10 hippocampal neurons from 3 independent experiments. $N_{\text{DRP1WT}} > 3000$, $N_{\text{DRP1K38A}} > 3000$. Kolmogorov–Smirnov test: $p = 3,5 \times 10^{-4}$.

(e) Quantification of the sum of lengths of mitochondria longer than 10 μm divided by the length of all mitochondria, showing that the percentage of long mitochondria with respect to the total mitochondrial network is significantly higher in DRP1K38A with respect to DRP1WT. Grey dots are data averaged per cell: $N_{\text{DRP1WT}} = 15$, $N_{\text{DRP1K38A}} = 26$, from three independent experiments. The horizontal lines represent the means, the error bars represent the standard error of the means, and the colored diamonds represent the means per experiment.

(f) Same cells shown in C, where the mitochondrial network (ATPb immunostaining, magenta) is shown together with the peroxisomes (PEX14, yellow).

(g) Representative ROIs from panel F showing the decreased amount of peroxisomal structures detected in proximity with mitochondria upon expression of DRP1K38A with respect to the control cells (DRP1WT).

(h) Quantification of the linear number density of overlapping sites per mitochondrial length between peroxisomal punctae labeled with the pre-peroxisomal biogenesis markers PEX14 and PEX3 (right graph) or PMP70 (left panel) and mitochondrial longer than 10 μm , measured in DRP1K38A or DRP1WT expressing cells.

Minor Concerns:

1. The authors should include novelty and motivation for using neuronal cultures in the background, as suggested by Reviewer 2. This would differentiate the study and support their claims of neuronal novelty.

The “main” section has been adjusted to include novelty and motivation for the importance of studying MDVs in neuronal cells.

2. Reviewer 3’s comments state that size cannot be used alone to determine an MDV, and the authors agree with the comments. However, the text was not altered to address these changes. Lines 121-123 still suggest using size to define an MDV.

The text has been modified according to the reviewer’s comment, now using mitochondrial-derived structures (MDSs) as a classification of all size-filtered nano-sized mitochondrial-derived structures.

3. The authors state that “only those attached to mitochondria and labelled with OMP25 were considered” for quantification in Figure 8. Could the authors state what percentage of vesicles were used in this quantification? If it is a small population, this is important to know for the interpretation

of the data. It is unclear whether this minor concern was addressed based on the revised manuscript.

The choice of considering only mitochondrial protrusions that remain connected to the mitochondrial tubule is necessary to limit the observation to the very initial stage of peroxisome *de novo* biogenesis and to better access the localization of ER and mitochondrial derived pre-peroxisome markers. Including the whole population of MDVs in the analysis would likely obscure the specific feature we aim to see.

4. There are no error bars on the normalized frequency plots (example, Fig. 4G).

The normalized frequency plots in Fig. 4g, Fig. 6g and Fig. 8 f-g were adjusted according to the reviewer suggestions and were substituted with box plots.

Reviewer #2 (Remarks to the Author):

The authors have done a fair job in addressing my previous comments, except for specific comment No. 4 (Fig. 1B, line profile #2: the overall confocal profile appears to be narrower than the STED profile, which shouldn't be the case.), which they could address with finding/acquiring pairs of images (confocal vs. STED) where the potential confounder of structural movement is absent. Otherwise, I find the paper suitable for Nat. Communications.

We thank the reviewer for the work done and the positive response.

We followed the reviewer advice and updated Fig. 1. Specifically, we replaced the second pair of images (from the left) in Fig. 1b, which were potentially affected by structural movements, with a new pair of images from a ROI where structural movements between STED and confocal are absent. The new pair of images shows two separate MDVs (FWHM1 = 138 nm; FWHM2 = 106 nm) detected in STED but not in the confocal comparison, where it looks like there is only one big vesicle (FWHM = 356 nm), due to the proximity of the two vesicles and the lack of resolution in the confocal scanning.

Reviewer #4 (Remarks to the Author):

The concerns have been addressed, and the manuscript is well organized for publication. We thank the reviewer for the work done and the positive response.

Reviewer #5 (Remarks to the Author):

I co-reviewed this manuscript with one of the reviewers who provided the listed reports. This is part of the Nature Communications initiative to facilitate training in peer review and to provide appropriate recognition for Early Career Researchers who co-review manuscripts. We thank the reviewer for the work done.

Reviewer #6 (Remarks to the Author):

NCOMMS-24-82932-T

We thank the reviewer for the work done.

Point by point response to Reviewers of the manuscript NCOMMS-24-82932A:

Quantitative optical nanoscopy of mitochondrial-derived vesicles in neurons classify pre-peroxisomal and clearing organelles
Coceano et al.

We thank the editor for the interest in our work and the reviewer for the final critical assessment of our study.

We revised the paper to address the editorial requests (following the Author Checklist) and we addressed the last reviewer questions.

Please find below a point-to-point response (blue) to the reviewers' questions (black).

Minor Comments:

- In Figure 9e-f, the authors mention "nearby" ER signal as part of their quantification criteria, but I do not see where this characteristic was defined. Was there a specific distance that the ER signal needed to fall within?

We thank the reviewer for the comment. We updated the relative paragraph in the Methods section as following: "To access ER association, fluorescence intensity was measured both at the same position overlapping the protrusion and along the protrusion boundary facing the mitochondrial side. "

Basically, to evaluate if the ER was associated to the protrusion or not, we measured the fluorescence intensity both at the same position overlapping the protrusion and at the border of the protrusion facing the mitochondrial side.

- Occasionally, there are grammatical errors in the new text (for example, Line 387 says "this mutant exhibit reduced efficiency" instead of "this mutant exhibits reduced efficiency"; Line 217, the word few is misspelled), so I recommend a quick grammar check before finalizing the paper.

We thank the reviewer for the thorough check. We performed a grammar check of the manuscript and corrected the errors found.